# Echo Chamber: RL Post-training Amplifies Behaviors Learned in Pretraining

**Rosie Zhao**[*]
Harvard University
Kempner Institute

**Alexandru Meterez**[*]
Harvard University
Kempner Institute

**Sham Kakade**
Harvard University
Kempner Institute

**Cengiz Pehlevan**
Harvard University
Kempner Institute

**Samy Jelassi**[†]
Harvard University

**Eran Malach**[†]
Harvard University
Kempner Institute

## Abstract

Reinforcement learning (RL)-based fine-tuning has become a crucial step in post-training language models for advanced mathematical reasoning and coding. Following the success of frontier reasoning models, recent work has demonstrated that RL fine-tuning consistently improves performance, even in smaller-scale models; however, the underlying mechanisms driving these improvements are not well-understood. Understanding the effects of RL fine-tuning requires disentangling its interaction with pretraining data composition, hyperparameters, and model scale, but such problems are exacerbated by the lack of transparency regarding the training data used in many existing models. In this work, we present a systematic end-to-end study of RL fine-tuning for mathematical reasoning by training models entirely from scratch on different mixtures of fully open datasets. We investigate the effects of various RL fine-tuning algorithms (PPO, GRPO, and Expert Iteration) across models of different scales. Our study reveals that RL algorithms consistently converge towards a dominant output distribution, amplifying patterns in the pretraining data. We also find that models of different scales trained on the same data mixture will converge to distinct output distributions, suggesting that there are scale-dependent biases in model generalization. Moreover, we find that RL post-training on simpler questions can lead to performance gains on harder ones, indicating that certain reasoning capabilities generalize across tasks. Our findings show that small-scale proxies in controlled settings can elicit interesting insights regarding the role of RL in shaping language model behavior.[1]

## 1 Introduction

Reinforcement learning-based fine-tuning has emerged as a crucial step in the post-training process for enhancing language models' capabilities in advanced mathematical reasoning and coding (Jaech et al., 2024; Guo et al., 2025; Shao et al., 2024; Team et al., 2025). Open-source efforts to reproduce the fine-tuning strategies used in state-of-the-art reasoning models have further demonstrated that reinforcement learning consistently boosts performance in these domains (Lambert et al., 2024; Havrilla et al., 2024; Luo et al., 2025; Zeng et al., 2025), even when applied to smaller-scale pretrained models or synthetic environments (Pan et al., 2025).

---

[*]Equal contribution. Correspondence to Rosie Zhao (`rosiezhao@g.harvard.edu`) and Alexandru Meterez (`ameterez@g.harvard.edu`).

[†]Equal contribution.

[1]Our code is available at `https://github.com/rosieyzh/openrlhf-pretrain`. All pretrained base models can be found here, and intermediate checkpoints from RL fine-tuning for two 1B pretrained models can be found at the following links: TinyGSM + OMI1 + OMI2 and TinyGSM + OMI2.

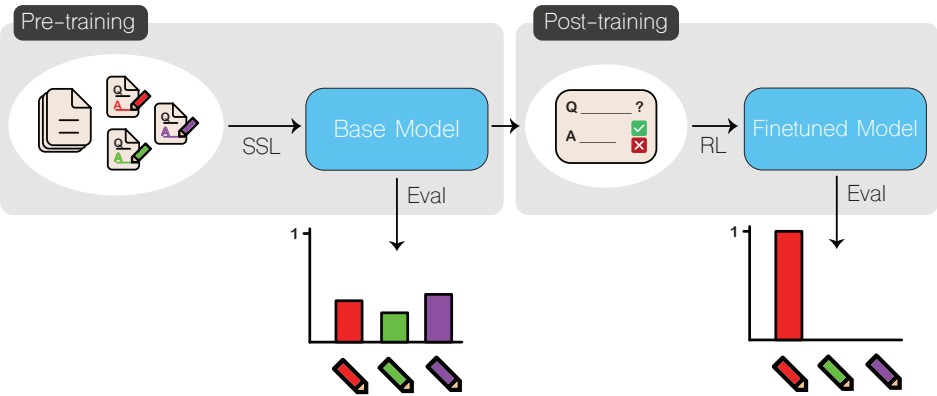

Figure 1: We conduct a systematic end-to-end study of RL fine-tuning for mathematical reasoning by training models entirely from scratch using different mixtures of datasets. The instruction datasets included in our pretraining mixes contain distinct formats which we can track in the model's generations after pretraining and RL post-training; we find that after post-training, the model consistently converges to a dominant output distribution coinciding with a significant increase in performance.

While RL post-training has demonstrated empirical success, the underlying mechanisms driving these improvements are being actively studied. Several hypotheses have been proposed to explain the effectiveness of RL, including its potential to encourage longer chains of thought (Wei et al., 2022; Yeo et al., 2025), facilitate backtracking behaviors (Guo et al., 2025), generalize to unseen task variants (Chu et al., 2025), and improve overall reasoning accuracy. However, a limitation of these studies is their lack of control over the pretraining data—an increasingly recognized factor in providing the proper model initialization needed for effective fine-tuning (Abdin et al., 2024; Allal et al., 2025; Petty et al., 2024; Penedo et al., 2024). This gap is especially salient given that most existing reproductions and analyses begin from base models whose pretraining datasets are either proprietary or insufficiently documented. A prominent example is the Qwen family of models (Yang et al., 2024), which is commonly used in RL post-training studies but the synthetic math and code data used for pretraining remains undisclosed. Prior work has shown that some models demonstrate substantial improvements while others stagnate when applying these post-training techniques (Gandhi et al., 2025), highlighting the critical influence of pretraining data— despite it being the most opaque part of the training pipeline for reasoning models. Consequently, it is difficult to isolate the role of RL in shaping model behavior, as its effects are entangled with unknown factors in the pretraining data.

In this work, we seek to clarify the relationship between pretraining data and RL-based post-training. Specifically, we ask the following: how does the composition of pretraining data affect the efficacy of RL fine-tuning? And how does this interaction depend on the choice of RL algorithm, the choice of hyperparameters, and model scale? To answer these questions, we construct a controlled experimental setting that allows us to systematically examine these factors, providing a clearer picture of how pretraining and RL jointly shape model behavior.

To isolate the effects of RL fine-tuning, we pretrain language models *from scratch* on curated mixtures of open-source datasets, including both document-style corpora and synthetic instruction datasets with diverse characteristics. This setup gives us full control over what the model is exposed to during pretraining and allows us to track the influence of specific instruction datasets. We then fine-tune these models using reinforcement learning on mathematical question-answering tasks. This controlled setting enables us to monitor both quantitative and qualitative shifts in the model's generations across different stages of training, offering a clearer view into the mechanisms by which RL fine-tuning interacts with pretraining data.

Our primary contributions are as follows:

- We conduct a principled investigation of RL fine-tuning starting from models of various scales that we have pretrained from scratch on mixtures of fully open datasets (Section 2).

- We find that RL fine-tuning consistently drives models to converge on generating outputs in the format of a single pretraining distribution (Section 3.1), often yielding improved pass@1 accuracy but reduced diversity. Despite occasional failure cases (Section 3.2), the preferred distribution is typically the most performant one - as measured on the base model's accuracy restricted to the specific distribution. Qualitative properties within the preferred distribution are also further refined during RL fine-tuning (Section 3.3).

- The preferred distribution reveals a scale-dependent bias: smaller models favor simpler, code-like formats, while larger models shift toward natural language outputs (Section 3.4).

- We provide evidence of positive transfer from RL fine-tuning, showing that models improve on evaluation datasets not seen during post-training (Section 4).

## 2 Experimental Setup

### 2.1 Pretraining

**Architecture:** We train decoder-only language models using the OLMo codebase (Groeneveld et al., 2024; OLMo et al., 2024) of two sizes: 150M and 1B parameters. The models have widths of 768 and 2048, and depths of 12 and 16 layers respectively. The MLP hidden dimension is 8x of the width, and we use SwiGLU activations (Shazeer, 2020) and RoPE positional encodings (Su et al., 2024).

**Datasets:** We train on a mixture of datasets related to mathematics; for all models, unless otherwise specified we train on FineMath-3+ (Allal et al., 2025) and the Algebraic-Stack subset of the Proof-Pile-2 (Azerbayev et al., 2023). Aside from these datasets consisting of documents with mathematical content, we also train on instruction datasets such as TinyGSM (Liu et al., 2023), OpenMathInstruct1 (Toshniwal et al., 2025b), and OpenMathInstruct2 (Toshniwal et al., 2025a). We repeat these question-answer datasets in various ratios in our mixtures, sometimes with multiple passes over the same dataset — we denote this using the $\times$ symbol throughout the manuscript (eg. $4\times$ TinyGSM refers to four passes over the TinyGSM dataset). We pretrain on the question-answer datasets by concatenating the prompt and the answer and adding them to the general corpus, without any chat template or special formatting.

TinyGSM is a synthetic dataset of 12.3M problem-solution pairs generated from the GSM8K and GSM-IC (Shi et al., 2023) training subsets, with code solutions generated by GPT-3.5. OpenMathInstruct1 consists of 1.8M problem-solution pairs generated from the GSM8K and MATH training subsets, with code solutions generated by Mixtral-8x7B (Jiang et al., 2024). Finally, OpenMathInstruct2 consists of 14M problem-solution pairs also generated from the GSM8K and MATH training subsets, with natural language solutions generated by Llama3.1-405B-Instruct. We focus on these datasets because each has distinct characteristics—such as tags and specific formatting—that we can search within the model's generations, enabling us to monitor the presence of each dataset throughout training. We provide more details and representative examples from each dataset in Appendix B.

**Pretraining Hyperparameters:** For all models we use the AdamW optimizer (Kingma & Ba, 2014; Loshchilov & Hutter, 2017) with a learning rate of 0.001 and weight decay of 0.1. We use a linear warmup of 5000 steps and a cosine decay scheduler to 10% of the peak learning rate.

## 2.2 Reinforcement Learning Fine-tuning

We perform fine-tuning using various RL algorithms directly on the models that we have pretrained from scratch. We use the OpenRLHF (Hu et al., 2024) implementation of Policy Optimization (PPO) (Schulman et al., 2017) and Group Relative Policy Optimization (GRPO) (Shao et al., 2024). We train using verifiable rewards (Lambert et al., 2024), where the reward function for RL fine-tuning is 1 if the model's answer matches the ground truth, and 0 otherwise.

We additionally fine-tune our models with Expert Iteration (EI) (Anthony et al., 2017). Starting from our pretrained models, we generate $k = 64$ generations for each problem in the train set of GSM8K, and create a de-duplicated dataset of the generations which lead to a correct answer. We use this dataset to then perform supervised fine-tuning on the pretrained model. This procedure can be done in iterations, where the fine-tuned model from the previous iteration is used to generate the de-duplicated dataset of correct generations, and supervised fine-tuning is done on the **base** model.

For the results presented in Section 3 we fine-tune using questions from the train split of GSM8K and study the performance and format of the generations of the models on the test split of GSM8K, both during and after fine-tuning. In Section 4 we take the models fine-tuned using questions from GSM8K and evaluate on the test set of MATH-500 and AIME 1983-2024. In Appendix I we also perform PPO on questions from the train split of MATH. For more details about the hyperparameters used, refer to Appendix C.

## 3 RL on Models Pretrained from Scratch with Different Mixtures

In this section, we present a summary of our results after applying reinforcement learning fine-tuning using problems from GSM8K on our models which were pretrained from scratch. With the exception of a few results in Section 3.3, we always include FineMath3+ and Algebraic-Stack in our pretraining mixtures, and vary quantities of TinyGSM, OpenMathInstruct1, and OpenMathInstruct2. Furthermore, unless otherwise specified, figures in this section correspond to our runs with PPO on models with 150M parameters; we conduct further analysis on models with 1B parameters in Section 3.4 and Appendix E, and comparisons with other RL algorithms and Expert Iteration are provided in Section 3.5 and Appendix F. Finally, we provide a brief theoretical justification of our results in Section 3.6.

### 3.1 RL converges to favour one distribution in the mixture

We begin by highlighting a striking pattern consistently observed during RL fine-tuning across all pretraining data mixtures: the model rapidly converges to producing outputs that follow the format of a single data distribution seen during pretraining, supressing the other ones. In Figure 2, we illustrate both the percentage of generations corresponding to each dataset and their respective accuracies when fine-tuning a model pretrained on TinyGSM, OpenMathInstruct1, and OpenMathInstruct2. For more details on dataset examples, how we evaluate the correctness of model generations, and the metrics that we report, see Appendix B. The model quickly shifts toward generating answers in the format of one distribution—TinyGSM in this case—within the first epoch (note the log-scaled x-axis). This transition coincides with the largest gain in overall pass@1 accuracy.

We also observe that while majority@64 accuracy improves by approximately 5% due to fine-tuning, pass@64 accuracy declines towards the end of training, in line with prior findings on reduced generation diversity following RLHF/RL fine-tuning (Kirk et al., 2024; Dang et al., 2025).

Additionally, we find that increasing the coefficient for the KL penalty during fine-tuning preserves some outputs in formats from other distributions besides the preferred one. As shown in Figure 3, fine-tuning with a higher KL coefficient for the same pretrained model from Figure 2 still results in a preference for TinyGSM-style outputs, but a subset of generations in natural language / OpenMathInstruct2 format still remains. This leads to a comparable pass@1 accuracy relative to the lower KL setting, while pass@64 accuracy

remains stable. In Appendix D, we demonstrate that this tendency to favor a single data distribution is consistent across all pretraining mixtures evaluated, and we also show that removing the KL penalty altogether yields similar performance.

Finally, although we focus on accuracy and percentage metrics for our analysis here and henceforth in this section, we show that similar phenomena manifest even when tracking confidence-based metrics— such as the average probability of the TinyGSM and OpenMathInstruct1-style initial token formats— in Appendix G.

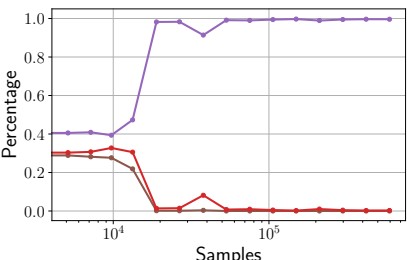 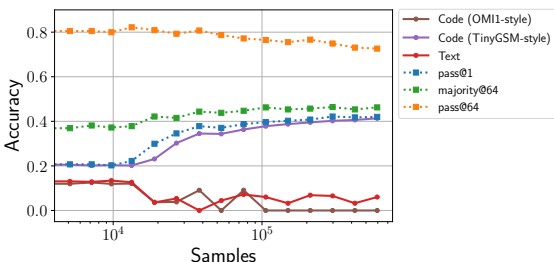

Figure 2: Starting from a 150M model pretrained with **TinyGSM, OpenMathInstruct1, and OpenMathInstruct2**, we track the following throughout PPO training: (**Left**) Percentage of generations on GSM8K test which adhere to the formats TinyGSM, OMI1, and Text (referring to the formats of TinyGSM, OpenMathInstruct1, and OpenMathInstruct2/natural language respectively) and (**Right**) GSM8K test accuracy restricted to the generations in each dataset format as well as overall pass@1, pass@64, and majority@64 accuracy. The generations quickly converge to outputting exclusively in the format of TinyGSM within the first epoch of training, which coincides with the greatest increase in overall accuracy. While majority@64 experiences a slight increase after fine-tuning, pass@64 performance decreases slightly at the end of training.

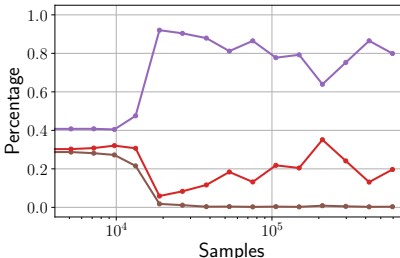 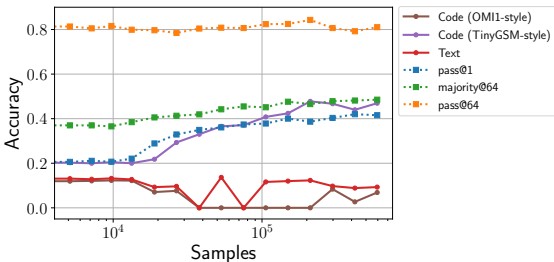

Figure 3: GSM8K test accuracy across epochs over the data during PPO when starting from the same 150M model as in Figure 2 but with a higher KL coefficient (0.01 as opposed to 0.001). The model still retains some generations using the format from OpenMathInstruct2, but reaches a similar final pass@1 accuracy as in Figure 2.

## 3.2  RL doesn't always favor the most performant, nor the most common distribution

In the previous section, we observed that RL fine-tuning amplifies generations coming from one distribution, while downweighting the others. This raises a natural question: does the model consistently favor the distribution that yields the best performance, or the distribution with the highest proportion of generations at initialization?

We find that the answer is nuanced and can depend on the pretraining data mixture. We provide two representative examples: in Figure 4, we present the evolution of the percentage of generations for each distribution and their accuracies during fine-tuning for models

pretrained on TinyGSM combined with varying amounts of OpenMathInstruct1. In Figure 4 (a), although the model initially produces more OpenMathInstruct1-style solutions (62%) compared to TinyGSM-style solutions (28%), it ultimately converges to generating TinyGSM-style outputs within the first epoch. In contrast, Figure 4 (b) shows that when the number of OpenMathInstruct1 samples is doubled during pretraining, the model instead converges to OpenMathInstruct1-style generations. This occurs despite the initial generation distribution being similar to Figure 4 (a) and despite TinyGSM generations achieving higher accuracy than OpenMathInstruct1 generations at initialization. However, in (b), the model achieves lower performance after fine-tuning compared to (a) and eventually degrades further near the end of training. We consider this a failure mode of RL fine-tuning. Nonetheless, in most of our experiments, the model tends to select the distribution with the highest performance after pretraining—TinyGSM, in the case of the 150M models—across the majority of fine-tuning runs.

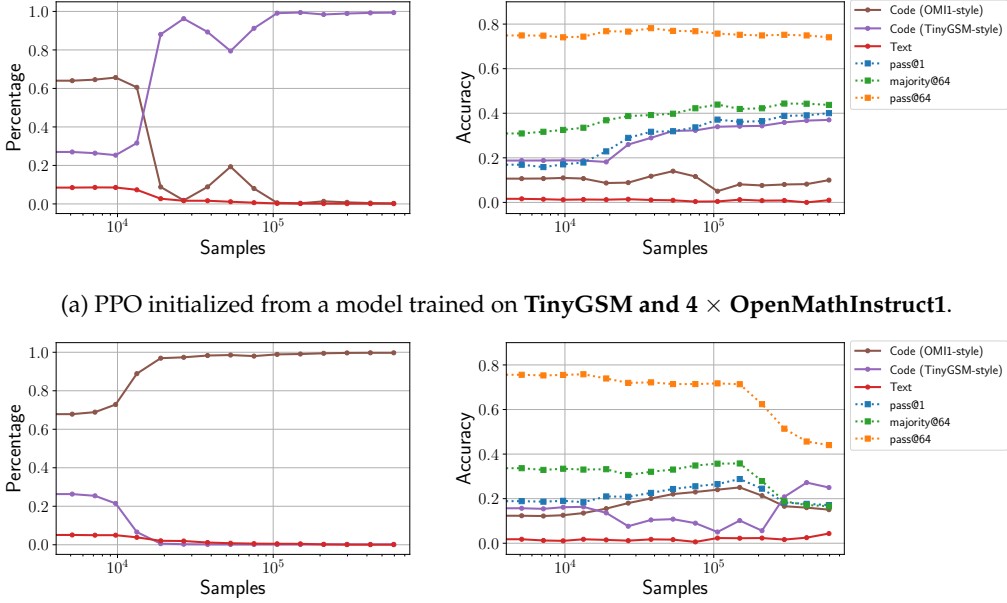

(a) PPO initialized from a model trained on **TinyGSM and 4 × OpenMathInstruct1**.

(b) PPO initialized from a model trained on **TinyGSM and 8 × OpenMathInstruct1**.

Figure 4: Proportion of generations by data format (left) and corresponding accuracies (right) during PPO fine-tuning with pretraining 150M models on TinyGSM and varying amounts of OpenMathInstruct1. In (a), where the pretraining set includes 4× OpenMathInstruct1, the model rapidly shifts within the first epoch to predominantly generating TinyGSM-style outputs, despite their lower frequency at initialization. In (b), increasing the amount of OpenMathInstruct1 in pretraining further results in the base model retaining a similar initial generation distribution. However, during fine-tuning, the model transitions to almost exclusively producing OpenMathInstruct1-style generations, which coincides with a drop in overall accuracy.

## 3.3   How does performance within one distribution improve during RL?

In the preceding sections, we examined models pretrained on varying proportions of the TinyGSM, OpenMathInstruct1, and OpenMathInstruct2 datasets (as a reminder, we always include FineMath3+ and Algebraic-Stack as well unless otherwise specified). We observed that, in most instances, the largest gains in pass@1 accuracy were associated with the model conforming to the format of a single distribution—in most cases, TinyGSM. This naturally raises the question of whether model generations exhibit meaningful progress *within* a given distribution, and whether performance improvements are achievable when pretraining is done on a single dataset.

Figure 5 (left) demonstrates that increasing the amount of TinyGSM data (specifically, we repeat TinyGSM 1, 2, 4, and 8 times in the pretraining mix) in the pretraining of 150M-parameter models leads to improved performance across pass@1, pass@64, and majority@64 accuracy after fine-tuning. Figure 5 (right) further illustrates the progression of pass@1 accuracy across training epochs, where we observe that models pretrained with the highest proportion of TinyGSM not only achieve the best final performance but also exhibit the largest performance gain from fine-tuning. We track the progression of pass@64 and majority@64 accuracy in Figure 10 in the Appendix. These findings suggest that selectively repeating subsets of pretraining data, rather than incorporating additional diverse datasets, may yield more substantial improvements due to RL-based fine-tuning.

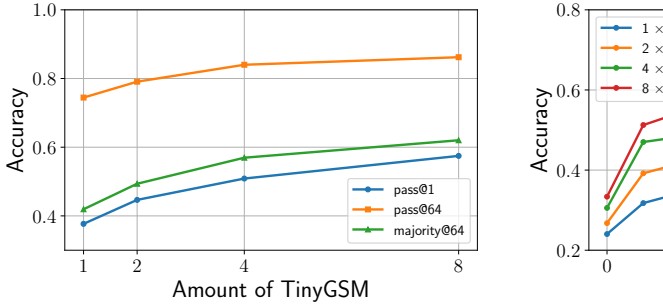

Figure 5: (Left): Top pass@1, pass@64, and majority@64 accuracy on GSM8K test across epochs after training with PPO on 150M models pretrained with **different amounts of TinyGSM**. (Right): GSM8K pass@1 test accuracy across PPO training for models trained on different amounts of TinyGSM.

Finally, we pretrain a 150M parameter model from scratch using **only TinyGSM, excluding FineMath3+ and Algebraic-Stack**. Our goal was to answer two questions: does RL fine-tuning still yield performance gains in the absence of additional datasets, and if so, what underlies these improvements?

As shown in Figure 6 (left), performance continues to improve after applying PPO to this model. To better understand how the model's generations evolve during fine-tuning, we track characteristic features of TinyGSM solutions — such as including a docstring that replicates the original question and having a lack of additional comments. In Figure 6 (right), we plot the proportion of model outputs that follow these conventions. We observe that, over training, the model increasingly conforms to the TinyGSM style, including settling on a consistent docstring format (e.g. shifting from mixed usage of single and double apostrophes to consistently using apostrophes). This supports the view that fine-tuning not only steers the model toward a preferred distribution but also refines outputs within that distribution. We further explore how fine-tuning improves generation quality beyond distributional preference in Section 4, where we discuss positive transfer effects to external evaluation datasets.

### 3.4 The effect of scale: larger models prefer different distributions

In this section, we examine how the trends identified above change with model scale. We pretrain 1B parameter models on various dataset mixtures to compare their behavior after fine-tuning with that of the corresponding 150M parameter model pretrained on the same mixture. We find that while models at both scales maintain a preference for a single distribution's format, the specific favored distribution changes with scale. Notably, 150M models tend to predominantly output TinyGSM-format generations, whereas the 1B models tend to prefer OpenMathInstruct2-style natural language responses, followed by OpenMathInstruct1-style code. As shown in Figure 7 and Appendix E, TinyGSM is not the preferred choice for the 1B models, and their final accuracy surpasses that of the smaller model pretrained on the same mixture. This points to a scale-dependent bias in

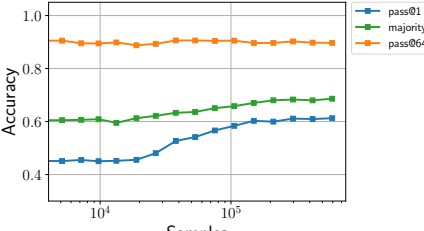 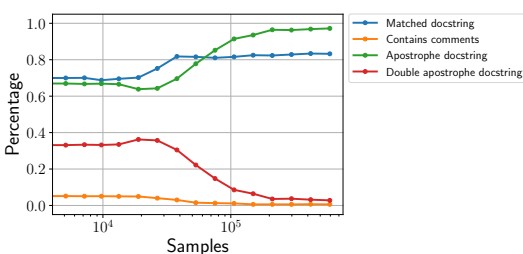

Figure 6: (Left): pass@1, pass@64, and majority@64 accuracies on the GSM8K test set during fine-tuning of a 150M model pretrained **solely with** $4\times$ **TinyGSM** (no Algebraic-Stack or FineMath3+). As with other pretraining mixtures, we continue to observe gains in final performance. (Right): Monitoring qualitative properties of the model's generations throughout fine-tuning, such as whether the docstring copies the question, the inclusion of comments, and the choice between single or double apostrophes for docstrings. The model progressively refines its outputs during training and increasingly aligns with the TinyGSM format, which coincides with improved accuracy.

behavior, likely tied to the larger model's greater capacity to answer questions correctly in natural language. In contrast, the 150M model may rely more heavily on the simpler, more deterministic TinyGSM-style code to produce accurate answers.

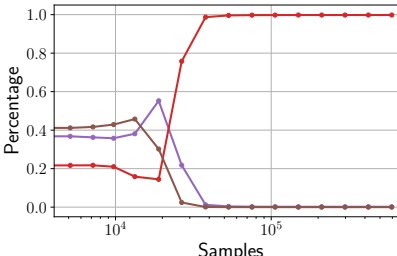 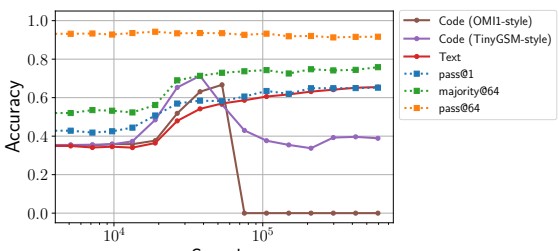

Figure 7: Percentage of generations (left) and respective accuracies (right) during PPO training for a 1B model pretrained on **TinyGSM, OpenMathInstruct1, and OpenMathInstruct2**. Although a 150M model pretrained on the exact same data converges on outputting only TinyGSM-formatted generations (see Figure 2), here we see the model amplify natural language solutions, even though natural language has the lowest percentage across generations and TinyGSM is the more performant distribution at initialization.

## 3.5 The effect of the RL algorithm

In Appendix F we report analogous results from the previous sections with GRPO and Expert Iteration. For GRPO in Appendix F.1 we observe the same trend in the percentage of generations where the model converges to favoring the format of one distribution, but the training of GRPO is generally less stable and often experiences a brief collapse in performance before recovering by the end of training. Additional results from multiple rounds of Expert Iteration are presented in Appendix F.2. In our setup, this approach consistently underperforms PPO and exhibits only a mild shift toward favoring a single dataset format. We believe this is likely due to repeatedly fine-tuning from the original base model. The nuanced differences we observe across RL algorithms highlight the need for further investigation into how specific algorithmic choices influence model behavior.

### 3.6 Supporting theory

We now provide some theoretical explanation for the results detailed above. We emphasize that the focus of this paper is not on theoretical analysis of reinforcement learning, and we simply reiterate known results that explain the findings of this work. Let $\mathcal{X}$ be the space of inputs and $\mathcal{Y}$ be the space of responses. Let $r : \mathcal{X} \times \mathcal{Y} \to \{0, 1\}$ be a reward function, and let $\pi_{\text{ref}}$ be our reference policy (before RL). Assume that our reference policy is in fact a mixture of $k$ different policies $\pi_1, \ldots, \pi_k$ s.t. $\pi_{\text{ref}}(y|x) = \sum_i \alpha_i \pi_i$, for $\alpha_1, \ldots, \alpha_k \in [0, 1]$ satisfying $\sum_i \alpha_i = 1$. For example, each $\pi_i$ can be a different solution format for math questions (code, text, etc.). We can frame the problem of reinforcement learning solved by e.g. PPO as maximizing the expected reward under KL-regularization[2]:

$$\arg\max_\pi \mathbb{E}_{y \sim \pi}[r(y, x)] - \frac{1}{\beta} \text{KL}(\pi, \pi_{\text{ref}})$$

Then, the maximizer would correspond to:

$$\pi^*(y|x) \propto \pi_{\text{ref}}(y|x) \exp\left(r(y, x)/\beta\right) = \sum_i \alpha_i \exp(r(x, y)/\beta) \pi_i(y|x)$$

Namely, we reweight the original mixture of policies corresponding to the rewards from each policy in the original mixture. This is consistent with our experiments, which show that RL mostly converges to the strategy which maximizes the reward.

## 4 Transfer to other evaluation datasets

In Section 3.3, we observed that RL fine-tuning can improve the structure of model outputs in ways that align with the format of the favored training distribution. While the qualitative attributes highlighted in Figure 6 may contribute to the model generating more accurate answers, our goal in this section is to gather stronger evidence that RL fine-tuning produces changes that directly enhance performance — such as reducing error rates or improving general capabilities like arithmetic. To this end, we focus on evaluating our models on datasets that were not used during fine-tuning, aiming to assess whether the models demonstrate positive transfer to more challenging tasks. For our 1B models, we evaluate on MATH-500 after performing PPO with the train questions from GSM8K and provide pass@1 and majority@64 performance before ('**Base**') and after ('**FT**') fine-tuning in Table 1. We observe consistent performance gains following fine-tuning, with some models improving by as much as 10%. Although MATH-500 is considered out-of-distribution relative to the fine-tuning data, models pretrained on mixtures that include either OpenMathInstruct datasets have already encountered synthetic problems resembling those in MATH. These models show the largest improvements on MATH-500 after fine-tuning, highlighting the benefit of pretraining on data that is structurally similar to the downstream task.

In Appendix H.1, we analyze these improvements from a qualitative lens by prompting GPT-4.5 Preview to classify the types of errors made by the base model for incorrect generations and later corrected following fine-tuning. In Appendix H.2 we present evaluation results on AIME for the same models and find little to no improvement on pass@1 and majority@64 performance for the AIME 2022–2024 benchmark across all pretrained models, but improvements are observed for pass@64 performance. In Appendix H.3 we provide examples of model generations on MATH-500 and AIME 2022-2024 before and after doing RL fine-tuning on GSM8K, where the base model was previously incorrect and the fine-tuned model provides a correct answer.

## 5 Discussion and Conclusion

In this work, we explored the effect of the pretraining data on the post-training stage in an end-to-end manner. Through pretraining models across different scales (150M and 1B) on

---

[2]We note that our experimental results hold even without adding the KL-regularization term. We leave an analysis of this setting to future work.

| Pretraining Data Mixture | Pass@1 Base | Pass@1 FT | Maj@64 Base | Maj@64 FT |
|---|---|---|---|---|
| TinyGSM + 4xOMI1 | 8.60% | 12.60% | 22.60% | 26.00% |
| TinyGSM + OMI2 | 33.40% | 43.60% | 46.20% | 52.80% |
| OMI2 + MMQA | 34.60% | 44.40% | 51.20% | 55.00% |
| TinyGSM | 4.80% | 9.60% | 7.80% | 12.20% |
| TinyGSM + OMI1 + OMI2 | 33.40% | 43.80% | 48.60% | 54.60% |

Table 1: Pass@1 and majority@64 performance of 1B models on the MATH-500 benchmark before and after RL fine-tuning with PPO on GSM8K train questions. Each row corresponds to a different pretraining data mixture. Results show consistent improvements after fine-tuning, suggesting that RL not only improves output formatting but also enhances general mathematical capabilities.

data mixtures containing general mathematics corpus and various ratios of question-answer datasets, our study has shown the following:

- RL fine-tuning amplifies a specific mode from the pretraining mixture while collapsing the others.

- The mode that gets amplified depends on the scale of the model, and the degree of amplification depends on the hyperparameters - namely, the coefficient for the KL penalty.

- RL post-training on simpler datasets such as GSM8K gives a performance boost on harder mathematical datasets such as MATH, and to a lesser extent on AIME.

- Small-scale proxies can offer valuable insights into the scientific aspects of RL fine-tuning in LLMs.

Our work opens up several exciting research directions towards understanding RL post-training and extracting more performance from these models. One potential question is how our results extend to more complicated data mixtures, such as including multilingual data in the mix. Moreover, is there a notion of an optimal pretraining mixture that would lead to the best reasoning performance downstream, and how does this mixture differ across model scales?

Crucially, we believe that one major confounder in the existing literature is the reliance on pretrained models. While several open-source reasoning models are openly available, the pretraining datasets are not public, which is a critical aspect of the performance of the base models on reasoning tasks (Yang et al., 2024; Grattafiori et al., 2024). Naturally, this discrepancy gets amplified in downstream fine-tuning and evaluation, leading to spurious conclusions about the abilities and behaviors of these models. We believe that studying LLM fine-tuning in controlled settings starting *from scratch* is a necessary and underexplored avenue for research, amenable for exploring in academic settings using the small scale proxies introduced in this manuscript.

# 6 Acknowledgements

SK, RZ, AM, and SJ acknowledge support from the Office of Naval Research under award N00014-22-1-2377 and the National Science Foundation Grant under award #IIS 2229881. This work has been made possible in part by a gift from the Chan Zuckerberg Initiative Foundation to establish the Kempner Institute for the Study of Natural and Artificial Intelligence. RZ is supported by a Simons Investigator Fellowship, NSF grant DMS-2134157, DARPA grant W911NF2010021,and DOE grant DE-SC0022199. CP is supported by NSF grant DMS-2134157, NSF CAREER Award IIS-2239780, DARPA grant DIAL-FP-038, a Sloan Research Fellowship, and The William F. Milton Fund from Harvard University. RZ and AM are supported by Kempner Institute Graduate Research Fellowships.

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

# A    Related Works

There is an extensive and rapidly expanding body of literature covering the understanding of post-training on the performance of LLMs in reasoning domains.

**Reasoning in Large Language Models:** Following the introduction of chain of thought (CoT) (Wei et al., 2022), LLMs have improved drastically in their reasoning capabilities. Frontier language models (Jaech et al., 2024; Grattafiori et al., 2024) have achieved impressive performance on hard mathematical and coding benchmarks (Hendrycks et al., 2021; Jain et al., 2024; Rein et al., 2024; Cobbe et al., 2021). Further lines of work expand upon the CoT concept towards more complex structures such as trees and graphs (Yao et al., 2023a; Besta et al., 2024). Another approach to improve performance on reasoning tasks is by combining CoT approaches with tools (Schick et al., 2023; He-Yueya et al., 2023; Yao et al., 2023b), or by teaching the model to produce formal representations - such as code, alongside the natural language generations (Guan et al., 2025). More recently, there have been several works proposing reasoning in latent thoughts, using different amounts of thinking tokens at training time and inference time (Hao et al., 2024; Zelikman et al., 2024).

**Reinforcement Learning Fine-tuning:** The post-training stage has been shown to be a crucial step towards improving LLM reasoning. Broadly, these can be split in supervised fine-tuning approaches (SFT)— which involve fine-tuning on a dataset, or distilling from a teacher model (Muennighoff et al., 2025; Xu et al., 2025)—, Expert Iteration (EI) approaches— usually involving training on multiple rounds on correct samples generated by the policy itself (Anthony et al., 2017; Dong et al., 2023; Gulcehre et al., 2023; Zelikman et al., 2022)—, and RL approaches—based on using a policy optimization algorithm (Schulman et al., 2017; Guo et al., 2025; Yu et al., 2025; Liu et al., 2025; Hu, 2025; Ahmadian et al., 2024; Kazemnejad et al., 2024). Recently, reinforcement learning with verifiable rewards (RLVR) (Lambert et al., 2024) has become the de facto standard for improving reasoning in LLMs, especially in mathematics and coding domains. In the case of reinforcement learning from human feedback (RLHF) for aligning models to human preferences, a reward model (Uesato et al., 2022; Lightman et al., 2023; Rafailov et al., 2023) is employed in order to rank the answers of the model to a prompt either at the end of the generation - termed outcome reward models (ORMs) (Cobbe et al., 2021), or at each intermediate step - termed process reward models (PRMs) (Cui et al., 2025; Yuan et al., 2024).

Despite the large literature covering RL post-training, there is still a lack of understanding for the connection between the pretraining data and the effect it has on RL post-training optimization. To the best of our knowledge, we are the first to perform an extensive end-to-end study of the effect of pretraining data mixtures for mathematical reasoning in LLMs of different scales, and explore the difference between the common policy optimization algorithms. A theoretical explanation for the diversity collapse brought by RLVR is presented in Wu et al. (2025), who argue that RLVR is inherently limited to the support of the base model. Havrilla et al. (2024) is the closest work to our own, studying the performance of PPO across scales both on base models and fine-tuned models. Pan et al. (2025) also explores the emergence of the "Aha" moment in base LLMs, trained for solving countdown and multiplication tasks. Finally, Gandhi et al. (2025) leverage continued pretraining on Llama models towards bringing their performance closer to the Qwen models, and show that this improvement correlates with the reasoning abilities of the initial model.

# B    Dataset and Evaluation Details

As mentioned in Section 2.1, we include TinyGSM, OpenMathInstruct1, and OpenMathInstruct2 instruction datasets in the pretraining mixture. Each of these datasets have distinct characteristics that can be searched for in the model's generations. We provide more details for each dataset here.

## B.1 TinyGSM

In TinyGSM, answers are formatted as Python code enclosed within a function named simple_math_problem(). This function consistently ends with return result, where result represents the final numerical solution to the grade-school math problem. To identify model generations that follow the TinyGSM format in our experimental results, we search for the function signature def simple_math_problem():. To evaluate for correctness, we run the code within simple_math_problem(). Additionally, these solutions include a docstring that replicates the problem statement. We track these characteristics in our experimental analysis, as discussed in Section 3.3. Below, we provide a representative example of a question and its corresponding solution.

---

**Representative Question in TinyGSM**

Benjamin picked some oranges at the fruit stand that cost $0.75 each. When Benjamin reached the cash register, he realized he was $9 short of the total price, so his friend Mason funded the rest. If Benjamin had $18 on him, how many oranges did he buy?

---

**Representative Answer in TinyGSM**

```python
def simple_math_problem() -> int:
    '''
    Benjamin picked some oranges at the fruit stand that cost
    $0.75 each. When Benjamin reached the cash register, he
    realized he was $9 short of the total price, so his
    friend Mason funded the rest. If Benjamin had $18 on him,
    how many oranges did he buy?
    '''
    cost_per_orange = 0.75
    amount_short = 9
    benjamin_money = 18

    total_cost = benjamin_money + amount_short
    number_of_oranges = total_cost / cost_per_orange

    result = number_of_oranges
    return result
```

## B.2 OpenMathInstruct1

In OpenMathInstruct1, answers are structured with code wrapped within `<llm-code></llm-code>` tags. Additionally, the parsed numerical result is enclosed in `<llm-code-output></llm-code-output>` tags, followed by a final boxed answer. For GSM8K evaluations, we execute the model-generated code within the `<llm-code></llm-code>` tags to assess correctness. In the case of MATH, since models may post-process the code output, we evaluate correctness based on either the executed code and the final boxed result. To identify model generations in our experimental results that adhere to the OpenMathInstruct1 format, we search for the presence of `<llm-code></llm-code>` tags. A representative question and answer is given below.

---

**Representative Question from OpenMathInstruct1**

Martha has 18 crayons. She lost half of them, so she bought a new set of 20 crayons. How many crayons in total does Martha have after the purchase?

---

**Representative Answer from OpenMathInstruct1**

Let's solve this problem using Python code.
```
<llm-code>

amount_of_lost_crayons = 18 / 2
amount_of_new_crayons = 20
total_amount = amount_of_lost_crayons + amount_of_new_crayons
total_amount

</llm-code>
<llm-code-output>
```
**Output:** `29.0`
```
</llm-code-output>
```
Thus, Martha has $\boxed{29}$ crayons in total.

---

### B.3 OpenMathInstruct2

In OpenMathInstruct2, responses are provided in natural language with a final boxed answer. Since this is the only dataset we consider that exclusively generates answers in natural language, if no Python code is detected in the specified TinyGSM or OpenMath-Instruct1 format, we classify the model's output as 'Text' and assume it likely follows the OpenMathInstruct2 format.

---

**Representative Question from OpenMathInstruct2**

A bag contains 8 red marbles and an unknown number of blue marbles. A player randomly selects a marble from the bag. If a red marble is selected, the player wins 3 dollars, but if a blue marble is selected, the player loses 3 dollars. If the expected amount won for playing the game is 1 dollar, then how many blue marbles are in the bag?

---

**Representative Answer from OpenMathInstruct2**

Let $b$ be the number of blue marbles in the bag.
The total number of marbles in the bag is $8 + b$.
The probability of selecting a red marble is $\frac{8}{8+b}$.
The probability of selecting a blue marble is $\frac{b}{8+b}$.
The expected amount won is calculated as the sum of the products of the amount won and the probability of each outcome:

$$\text{Expected amount won} = (3)\left(\frac{8}{8+b}\right) + (-3)\left(\frac{b}{8+b}\right)$$

We are given that the expected amount won is 1 dollar:

$$(3)\left(\frac{8}{8+b}\right) + (-3)\left(\frac{b}{8+b}\right) = 1$$

Simplify the equation:

$$\frac{24}{8+b} - \frac{3b}{8+b} = 1$$

Combine the fractions:

$$\frac{24 - 3b}{8+b} = 1$$

Multiply both sides by $8 + b$:

$$24 - 3b = 8 + b$$

Solve for $b$:

$$24 - 8 = 3b + b \Rightarrow 16 = 4b \Rightarrow b = \frac{16}{4} \Rightarrow b = \boxed{4}$$

---

### B.4 Evaluation

To evaluate model generations, we apply different procedures depending on the output format. If the model produces code—such as in the TinyGSM or OpenMathInstruct1 formats—we execute the code and extract the result: for TinyGSM, this is the value of the `result` variable, and for OpenMathInstruct1, it is the value of the variable on the last line within the `<llm-code></llm-code>` tags. Whether the model outputs code or natural language, the final answer is parsed using the Math-Verify library (Kydlíček, 2025) from HuggingFace to determine whether the prediction matches the correct answer.

We report three overall accuracy metrics: pass@1, pass@64, and majority@64. Pass@1 measures the percentage of questions correctly answered with a single generation using greedy decoding. Pass@64 reflects the percentage of problems for which at least one out of 64 sampled generations using temperature 0.7 produces a correct answer. Majority@64 measures the percentage of questions for which the most frequent final answer across 64 generations using temperature 0.7 matches the correct solution.

## C  Additional Experimental Details

We use the OpenRLHF (Hu et al., 2024) implementation of PPO and GRPO. The default hyperparameter configurations we use for these algorithms are in Table 2. We also vary KL coefficient to be 0 or 0.01. Other hyperparameters are set as default from OpenRLHF; for instance, for PPO we use the token-level KL penalty which is added to the reward, and for GRPO we incorporate the KL penalty in the loss and use the non-negative 'k3' estimator. We also use the hyperparameters in Table 3 for Expert Iteration (EI) results in Appendix F.2, where $k = 64$ is the number of samples we generate per problem before checking for correctness and filtering. We swept over peak learning rate values in $[5 \times 10^{-6}, 1 \times 10^{-5}, 1 \times 10^{-4}, 0.001]$ and observed very marginal gains (1-2%) for other learning rates in the first iteration of EI aside from $1 \times 10^{-4}$.

| Parameter | Value |
|---|---|
| Training Batch Size | 64 |
| Epochs | 10 |
| Prompt Max Length | 1024 |
| Generate Max Length | 1024 |
| Actor Learning Rate | $1 \times 10^{-6}$ |
| Critic Learning Rate | $7 \times 10^{-6}$ |
| Temperature | 0.7 |
| KL Coefficient | $1 \times 10^{-3}$ |
| Rollout Batch Size | 64 |
| Samples per Prompt | 8 |
| Reward Normalization | True |
| $\lambda$ | 0.95 |
| Clip $\epsilon$ | 0.2 |
| Warmup | 0.03 |
| Adam Betas | (0.9, 0.95) |

Table 2: Hyper-Parameter Configuration for PPO and GRPO runs.

| Parameter | Value |
|---|---|
| k | 64 |
| Training Batch Size | 256 |
| Epochs | 2 |
| Prompt Max Length | 1024 |
| Generate Max Length | 1024 |
| Learning Rate | $1 \times 10^{-4}$ |
| Adam Betas | (0.9, 0.95) |

Table 3: Hyper-Parameter Configuration for EI runs.

## D  Additional Mixtures - 150M Models

### D.1  Mixtures with OpenMathInstruct1 and OpenMathInstruct2

We provide additional results analogous to Figure 2 and Figure 3 for two other pretraining mixtures on our 150M models: TinyGSM and OpenMathInstruct1 (Figure 8) and TinyGSM

and OpenMathInstruct2 (Figure 9). As before, we also include FineMath3+ and Algebraic-Stack in the pretraining mixture. Across both mixtures we see the model converges to outputting TinyGSM-format code, with the exception of a high KL coefficient; we note in particular that for all of our mixtures, KL coefficient 0 yielded similarly performant results to the default setting 0.001, in line with prior work proposing to remove the KL penalty for fine-tuning reasoning models (Yu et al., 2025).

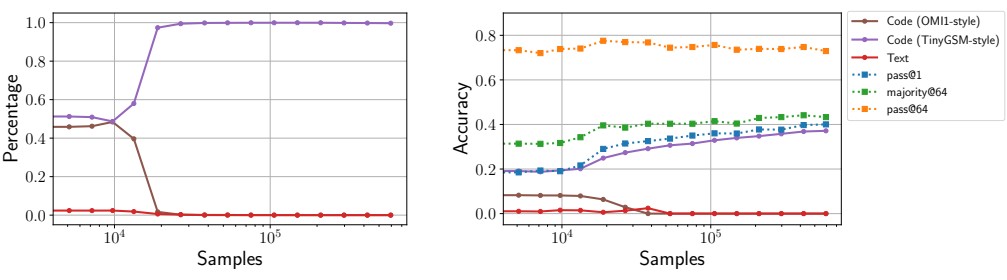

(a) PPO on a model trained on TinyGSM and $1\times$ OpenMathInstruct1 with KL coefficient 0.001.

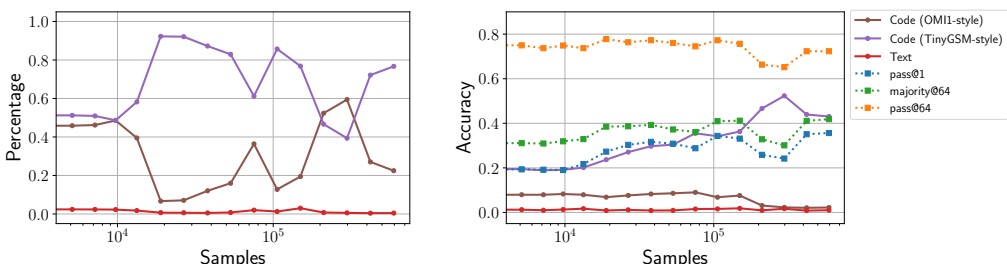

(b) PPO on a model trained on TinyGSM and $1\times$ OpenMathInstruct1 with KL coefficient 0.01.

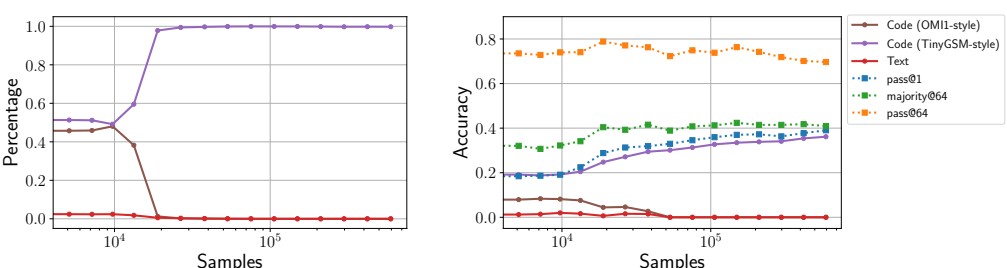

(c) PPO on a model trained on TinyGSM and $1\times$ OpenMathInstruct1 with KL coefficient 0.

Figure 8: Tracking percentage of generations and accuracy for PPO runs with varying KL coefficients, starting from a 150M model pretrained on **TinyGSM and OpenMathInstruct1**. We observe that TinyGSM is the consistently preferred distribution, and using KL coefficient 0 behaves similarly to KL coefficient 0.001.

## D.2 TinyGSM - Varying Fractions ($1\times$, $2\times$, $4\times$, $8\times$)

In Figure 10 we show how pass@64 and majority@64 performance progresses throughout PPO training starting from models pretrained on various amounts of TinyGSM (along with FineMath3+ and Algebraic-Stack). While majority@64 yields a 5-10% improvement across training, we note that pass@64 performance increases with the amount of TinyGSM shown in training but does not improve from model initialization during fine-tuning.

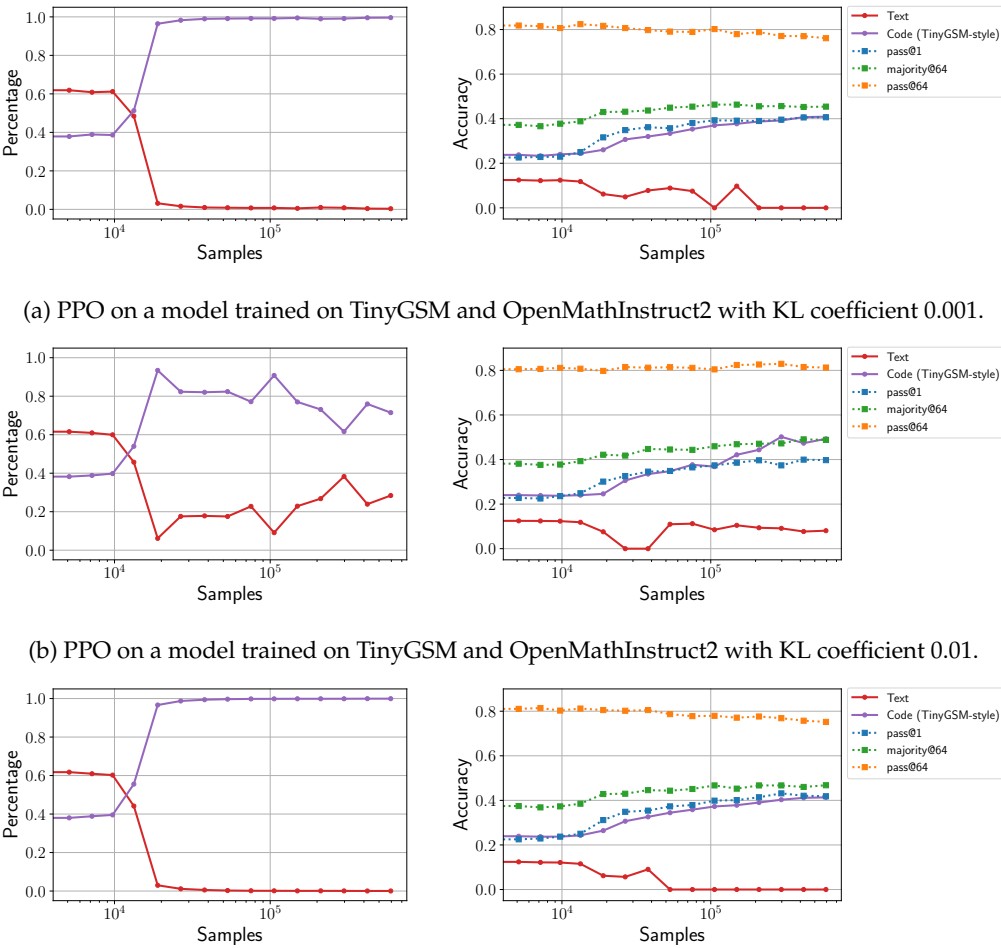

(a) PPO on a model trained on TinyGSM and OpenMathInstruct2 with KL coefficient 0.001.

(b) PPO on a model trained on TinyGSM and OpenMathInstruct2 with KL coefficient 0.01.

(c) PPO on a model trained on TinyGSM and OpenMathInstruct2 with KL coefficient 0.

Figure 9: Tracking percentage of generations and accuracy for PPO runs with varying KL coefficients, starting from a 150M model pretrained on **TinyGSM and OpenMathInstruct2**. We observe that TinyGSM is the consistently preferred distribution, and using KL coefficient 0 behaves similarly to KL coefficient 0.001.

# E    Additional Mixtures - 1B Models

Below we provide additional figures showing the percentage of generations and respective accuracies starting from 1B parameter models pretrained on different mixes of TinyGSM, OpenMathInstruct1, and OpenMathInstruct2. For all of our 1B models, we include the FineMath3+ and Algebraic-Stack datasets. In Figure 11 we perform PPO on a 1B model pretrained on TinyGSM and 4× OpenMathInstruct1 (corresponding 150M model shown in Figure 4(a)) and in Figure 12 we perform PPO on a 1B model pretrained on TinyGSM and OpenMathInstruct2 (corresponding 150M model shown in Figure 9(a)). We find that at this model scale, the model converges to outputting natural language rather than TinyGSM- or OpenMathInstruct1-style code. We also verify that mixing TinyGSM and OpenMath-Instruct2 yielded the highest performing model after fine-tuning, instead of having only TinyGSM or only OpenMathInstruct2 and MMQA in the pretraining mix (see Figure 13).

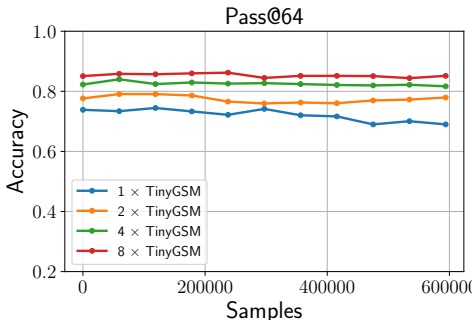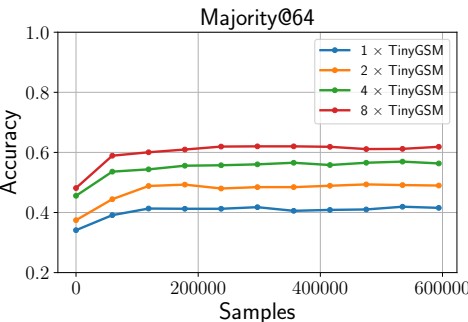

Figure 10: Pass@64 and majority@64 performance across epochs for the corresponding runs shown in Figure 5. While pass@k performance does not significantly improve after RL training, there is a 5-10% improvement in majority@k performance.

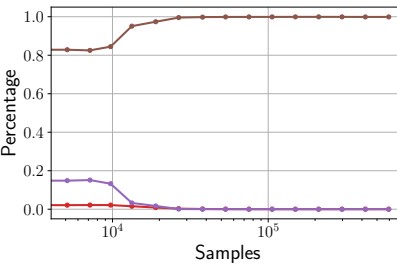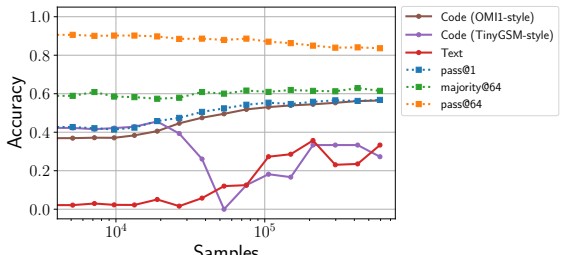

Figure 11: Percentage of generations (left) and respective accuracies (right) during PPO training for a 1B model pretrained on **TinyGSM and** $4\times$ **OpenMathInstruct1**. This is the same pretraining data used for the 150M model in Figure 4 (a), but here we see the 1B model amplify the OpenMathInstruct1 code format and obtaining a better final accuracy compared to the 150M model.

# F  Other RL Algorithms: GRPO, EI

## F.1  GRPO

We also perform RL fine-tuning using GRPO (Shao et al., 2024) using the same hyperparameters as for PPO. In Figure 14 we present analogous results for GRPO as Figure 2 and Figure 3 were for PPO. Across different data mixtures, we generally observed GRPO to exhibit the same phenomenon of preferring one distribution; however, it was less stable than PPO and often experienced a brief collapse in performance before recovering again by the end of training. In Figure 14, we see that the model switches its preference from natural language generations to TinyGSM, coinciding with this drop in performance. GRPO with a higher KL coefficient still exhibits the convergence to the TinyGSM format in contrast to PPO.

In Figure 15 we present analogous results as Figure 4 for GRPO. We see similar evolutions of the percentage of generations as in PPO, and the accuracy shows a similar collapse (in the case of training with $8\times$ OpenMathInstruct1, this model does not recover from this collapse).

Finally in Figure 16 we present analogous results as Figure 6 where we perform GRPO on a model trained on $4\times$ TinyGSM *only* (without Algebraic-Stack and FineMath3+) and in Figure 5 where we do GRPO on models trained on varying amounts of TinyGSM (with Algebraic-Stack and FineMath3+ included). We see that performance is very similar to PPO, with GRPO performing slightly worse for increasing amounts of TinyGSM in the pretraining data.

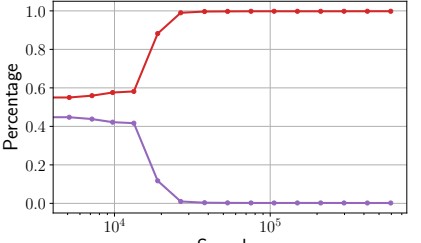 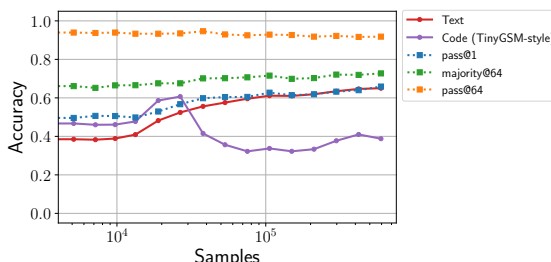

Figure 12: Percentage of generations (left) and respective accuracies (right) during PPO training for a 1B model pretrained on **TinyGSM and OpenMathInstruct2**. Although our 150M pretrained models most frequently converged on only outputting only TinyGSM-formatted generations, here we see the model amplify natural language solutions, even though TinyGSM is the more performant distribution at initialization.

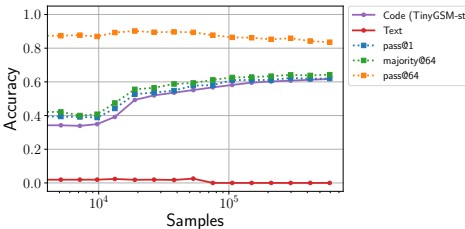 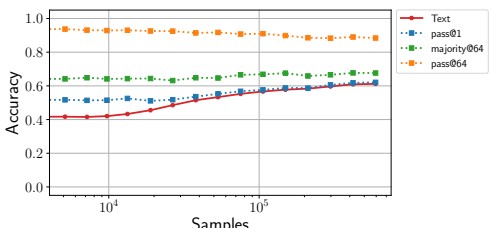

Figure 13: Accuracy during PPO training for a 1B model pretrained on **TinyGSM** (left) and on **OpenMathInstruct2 and MMQA** (right). For the 1B model on the left, its final accuracy is higher than the corresponding 150M model pretrained on the same amount of data (See Figure 5). However, both models trained on these subsets alone do not reach the same final accuracy as the model pretrained with the two datasets mixed (see Figure 12).

### F.2 Expert Iteration

We also ran Expert Iteration on a subset of our 150M pretrained models. As outlined in Section 2, we began by generating $k = 64$ candidate solutions per problem from the GSM8K training set using the pretrained model. From these, we constructed a de-duplicated dataset consisting only of generations that yield the correct final answer. This dataset was then used for supervised fine-tuning of the pretrained model. We repeated this process over multiple iterations: each time, the fine-tuned model was used to regenerate correct samples, while the training continued from the original base model. Our main goals were to assess whether one data format tends to dominate over others in the mixture and to compare performance against our PPO results, following similar questions posed in Havrilla et al. (2024). To ensure a comparable x-axis with our PPO results, we track the percentage and accuracy of generations as a function of the cumulative number of training samples. Specifically, for each iteration, we increment the total sample count by multiplying the number of training epochs with the size of the de-duplicated dataset.

In Figure 17, we present results from three iterations of Expert Iteration starting from the same 150M base model used in Figure 2, pretrained on a mixture of TinyGSM, OpenMathInstruct1, and OpenMathInstruct2. Despite seeing a comparable number of training samples, final performance lags behind that of PPO, and the model's generations do not show a strong preference for any particular dataset format. Nonetheless, there is a modest trend toward increased preference for TinyGSM over time, though this shift is slower and less pronounced; see Figure 18 and Figure 19 for similar experiments using base models pretrained on TinyGSM + OpenMathInstruct1 and TinyGSM + OpenMathInstruct2, respectively. Overall, we find that Expert Iteration consistently underperforms PPO—even in settings without dataset mixtures. For example, in Figure 20, starting from a base model pretrained

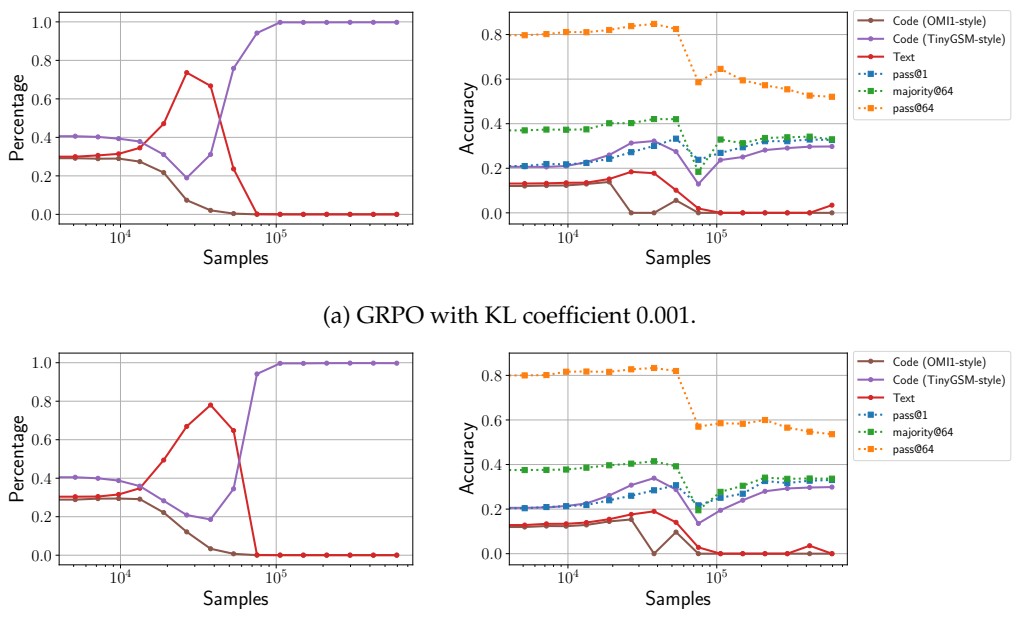

(a) GRPO with KL coefficient 0.001.

(b) GRPO with KL coefficient 0.01.

Figure 14: The analogous results using GRPO starting from the same model pretrained with TinyGSM, OpenMathInstruct1, and OpenMathInstruct2, with low KL ((a), analogous to Figure 2) and high KL coefficient ((b), analogous to Figure 3). GRPO exhibits less stable dynamics compared to PPO, where it appears that one distribution is about to be preferred but suddenly switches its preferences, corresponding with a drop in overall accuracy. Once the model has converged on one distribution, the accuracy begins recovering again. We also note that GRPO is more robust to high KL, likely due to the presence of the KL penalty in the loss as opposed to the reward (see Appendix C).

on $8\times$ TinyGSM (which achieves ~60% GSM8K test accuracy after PPO), accuracy after three EI iterations remains below 45%.

We also ran two iterations of EI on three of our pretrained 1B models. In Figure 21 observe similar trends where accuracy marginally improves and there is a modest trend towards an increased preference for OpenMathInstruct/natural language-style answers.

We hypothesize that the slower shift toward a dominant format is due to the repeated fine-tuning from the fixed base model, in contrast to PPO or GRPO's more online nature. This may suggest that more offline update steps in RL fine-tuning help maintain the original distribution, which could be beneficial for preserving generation diversity. We leave further exploration of RL algorithms and their associated design choices to future work.

## G   Confidence-Based Metrics

Our results in Section 3 highlight how different pretraining data mixtures influence both the stylistic distribution and accuracy of model outputs. We now show that these preferences also manifest in confidence-based metrics.

During RL fine-tuning, we track the average probability of outputs beginning with `def simple_math_problem()` and `Let's solve this problem using Python code. <llm-code>` on the GSM8K test set. As detailed in Appendix B, these token prefixes are characteristic of TinyGSM and OMI1-style generations, respectively. (We exclude OMI2 from this analysis due to the lack of a consistent initial token pattern.) As shown in Figure 22, the average probabilities closely follow the trends in output proportions presented in Figures 2, 4(a),

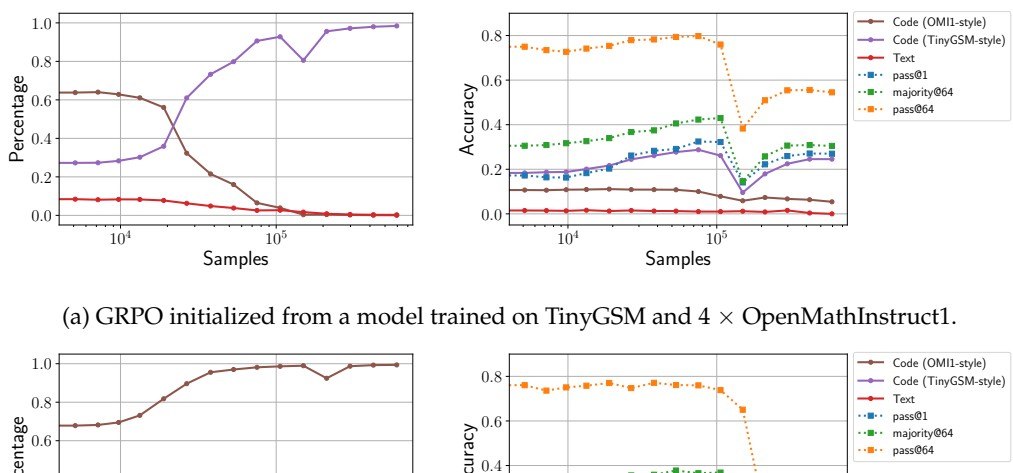

(a) GRPO initialized from a model trained on TinyGSM and $4 \times$ OpenMathInstruct1.

(b) GRPO initialized from a model trained on TinyGSM and $8 \times$ OpenMathInstruct1.

Figure 15: Analogous figure as Figure 4 when using GRPO instead of PPO. We see the same conclusion that TinyGSM is preferred in (a) and OpenMathInstruct1 is preferred in (b) which results in a collapse in performance. We observe the same initial increase and collapse later in training as mentioned in Figure 14.

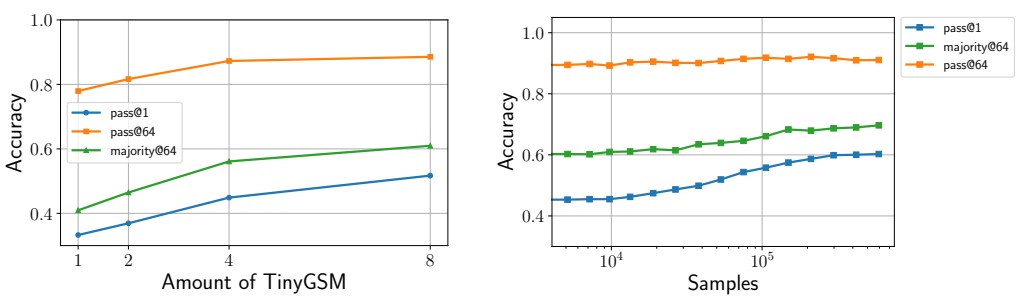

Figure 16: Analogous figures as Figure 5 (**Left** and Figure 6 (**Right**) ) when using GRPO instead of PPO. We see near-identical trends as in PPO, with the exception of pass@1 accuracy being slightly worse when increasing quantities of TinyGSM compared to PPO.

and 4(b), albeit with a smoother trajectory. Additionally, the narrowing error bars over the course of training suggest further stability.

Overall, we found that the average generation probabilities increase throughout training—even after the output format has largely stabilized—indicating that the model's confidence continues to grow within the dominant output distribution.

## H    Further Transfer Learning Investigations

### H.1    Qualitative Analysis on MATH-500 Generations

In Section 4, we demonstrated that 1B models fine-tuned on GSM8K questions showed improved performance on MATH-500. To further analyze these gains, for each of our models

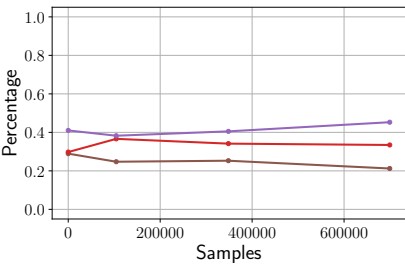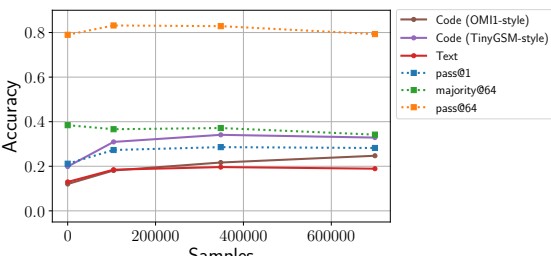

Figure 17: Percentage of generations (**Left**) and respective accuracies (**Right**) as a function of cumulative number of training samples for the same 150M model pretrained on **TinyGSM, OpenMathInstruct1, and OpenMathInstruct2**—as in Figure 2—across three iterations of EI. We note a lower increase in overall performance for roughly a similar number of examples for PPO, and the percentage of generations show only a slight preference for TinyGSM-format generations.

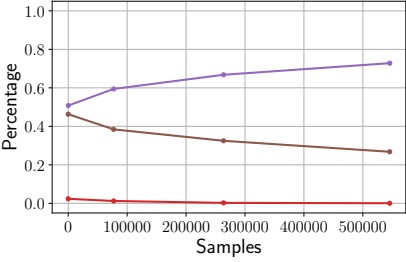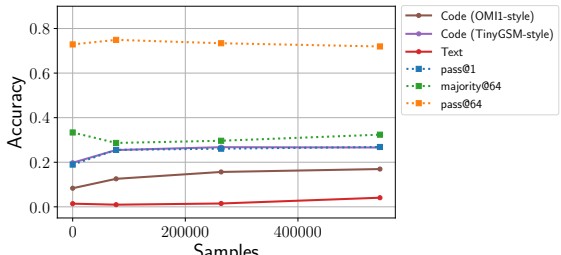

Figure 18: Percentage of generations (**Left**) and respective accuracies (**Right**) as a function of cumulative number of training samples for a 150M model pretrained on **TinyGSM and OpenMathInstruct1** across three iterations of EI. Here we see the final accuracy is lower than that of PPO (see Figure 8 (a)) and an increasing preference for TinyGSM.

we identified the subset of questions where the model's answer was initially incorrect after pretraining but became correct following fine-tuning. For each of these cases, we prompted GPT-4.5 Preview to explain why the base model's response was incorrect, why the fine-tuned model's response was correct, and to indicate which type of error was corrected between the two generations, from the following predefined set:

- **Arithmetic error** – Mistakes in calculation, sign, order of operations, rounding, or undefined operations.

- **Formula/application mistake** – Using the wrong formula, incorrect substitutions, or misapplying rules (e.g., differentiation, integration, exponentiation, trigonometry).

- **Algebraic/logic flaw** – Incorrect manipulation, missing/extra terms, or flawed reasoning in problem-solving.

- **Misinterpretation/misreading** – Incorrect understanding of the problem, assumptions, or misusing given information.

- **Notation/representation issue** – Errors in variables, indexing, units, graphing, or coordinate representation.

- **Incomplete answer** - Incorrect solution was incomplete or collapsed (started repeating, included irrelevant content, etc.)

Figure 24 presents a breakdown of error types made by each pretrained model, sorted in descending order from left to right. Across most models, the dominant sources of error stem from misinterpreting the question or making flawed algebraic or logical deductions.

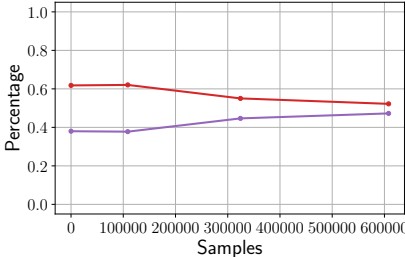 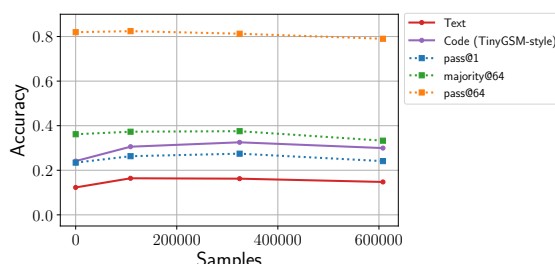

Figure 19: Percentage of generations (**Left**) and respective accuracies (**Right**) as a function of cumulative number of training samples for a 150M model pretrained on **TinyGSM and OpenMathInstruct2** across three iterations of EI. Here we see the final accuracy is lower than that of PPO (see Figure 9 (a)) with performance plateauing by the third iteration. We do see a similar trend as in Figure 9 (a) where TinyGSM-format code is starting to occupy a larger percentage of generations compared to natural language, but the effect is much slower compared to PPO.

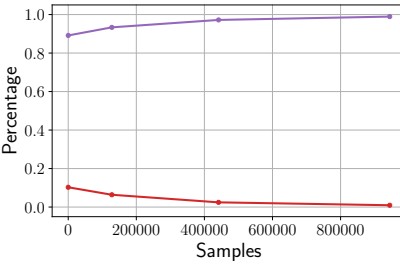 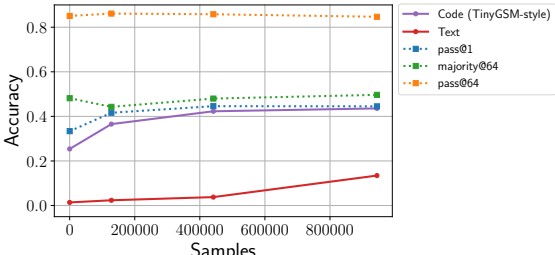

Figure 20: Percentage of generations (**Left**) and respective accuracies (**Right**) as a function of cumulative number of training samples for a 150M model pretrained on $8\times$ **TinyGSM** across three iterations of EI. After three iterations of EI, the model performance is below 45%, whereas after PPO the accuracy reaches almost 60%.

This suggests that the gains from fine-tuning are not driven by improvements just in better arithmetic accuracy. Instead, they appear to enhance the model's ability to comprehend the problem and reason through its solution, along with the format-level refinements discussed in Section 3.3.

## H.2 AIME Evaluation

In Section 4, we showed that evaluation on MATH-500 improved after applying PPO on GSM8K training questions. Here, we present additional evaluation results on AIME. As shown in Table 4, performance on AIME 2022–2024 questions exhibits minimal to no improvement in pass@1 and majority@64 metrics following PPO.

In contrast, Table 5, which includes a broader evaluation set spanning AIME 1983–2024, shows more substantial gains in both metrics. However, we do observe improvement in pass@64 performance for the two AIME subsets in Table 6. Notably, models pretrained on mixtures incorporating OpenMathInstruct datasets (which include synthetic problems derived from MATH) achieved the largest improvements after post-training. The observed pattern suggests that data similarity between pretraining and evaluation distributions is crucial for transfer. In particular, AIME questions prior to 2022 are known to have potential data contamination with MATH.

In Figure 23, we perform the same qualitative analysis on the generations for the AIME pass@64 evaluation as in Section H.1.

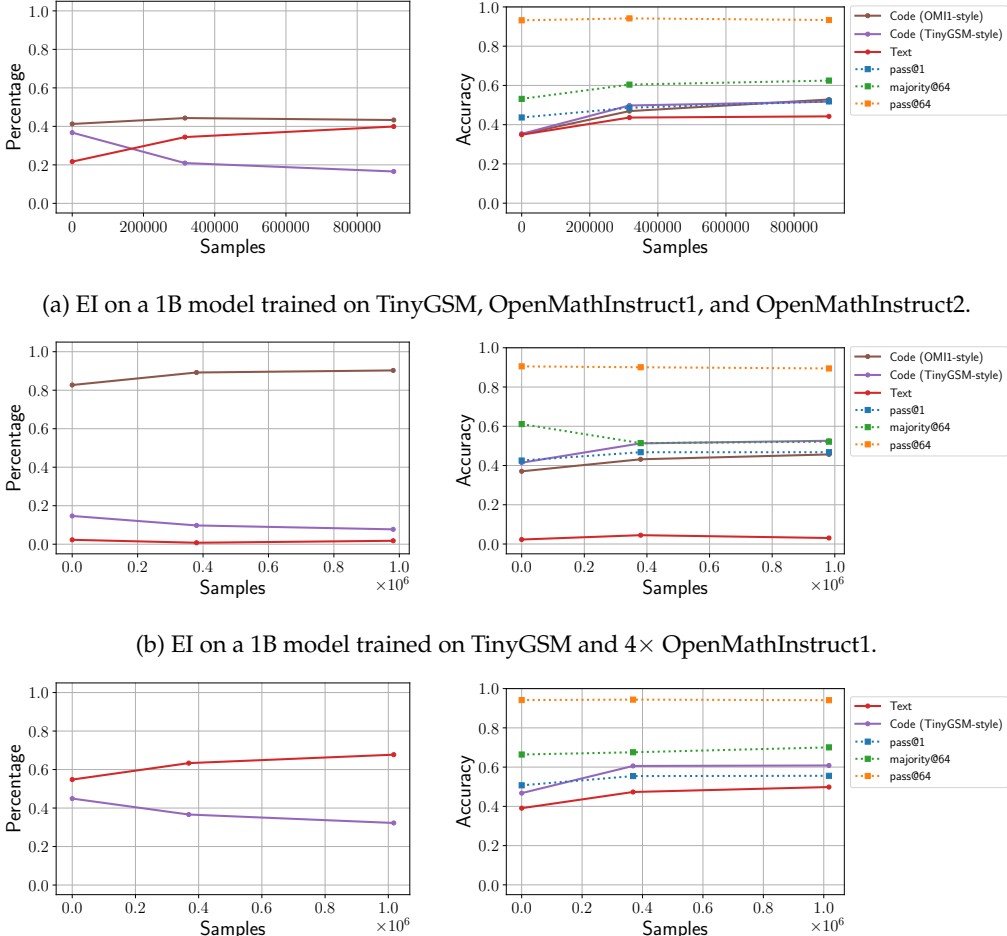

(a) EI on a 1B model trained on TinyGSM, OpenMathInstruct1, and OpenMathInstruct2.

(b) EI on a 1B model trained on TinyGSM and 4× OpenMathInstruct1.

(c) EI on a 1B model trained on TinyGSM, OpenMathInstruct2, and MMQA.

Figure 21: We perform two iterations of EI for starting from three 1B pretrained models. We see only a slight increase in overall performance, and a trend towards preferring natural language answers (consistent with our findings regarding the preferred distribution changing with scale in Section 3.4).

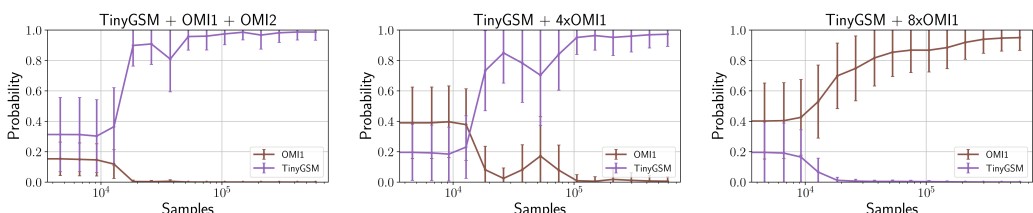

Figure 22: Average probability of def simple_math_problem() and Let's solve this problem using Python code. <llm-code> occurring after each problem in the GSM8k test set for models pretrained from **TinyGSM, OpenMathInstruct1, and OpenMathInstruct2** (left), **TinyGSM and** 4× **OpenMathInstruct1** (middle), and **TinyGSM and** 8× **OpenMath-Instruct1** (right). The average probability corresponding to generations from the preferred dataset in the percentage plots (from left to right, Figure 2, Figure 4(a), and Figure 4(b)) is similarly amplified over the course of RL fine-tuning.

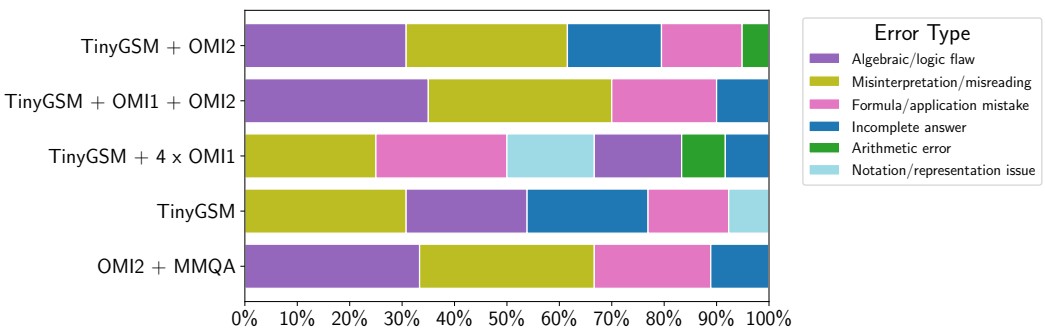

Figure 23: Distribution of error types on AIME for each 1B pretrained model before fine-tuning on GSM8K.

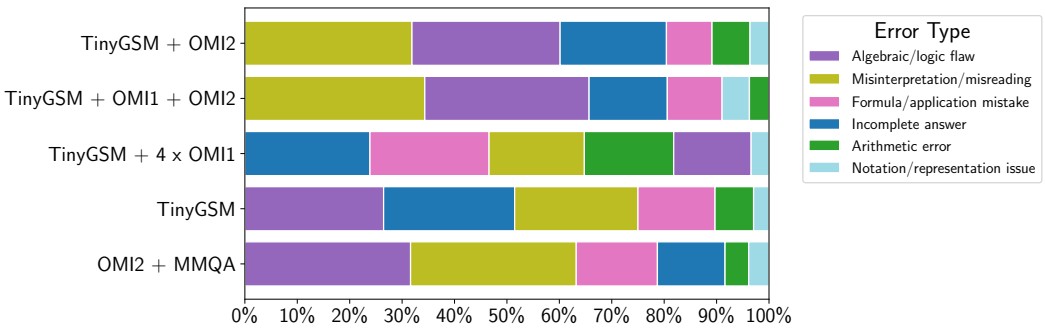

Figure 24: Distribution of error types on MATH-500 for each 1B pretrained model before fine-tuning on GSM8K.

| Pretraining Data Mixture | Pass@1 Base | Pass@1 FT | Maj@64 Base | Maj@64 FT |
|---|---|---|---|---|
| TinyGSM + 4xOMI1 | 0.00% | 0.00% | 0.00% | 0.00% |
| TinyGSM + OMI2 | 0.00% | 1.11% | 0.00% | 2.22% |
| OMI2 + MMQA | 1.11% | 2.22% | 1.11% | 3.33% |
| TinyGSM | 0.00% | 0.00% | 0.00% | 1.11% |
| TinyGSM + OMI1 + OMI2 | 0.00% | 2.22% | 1.11% | 2.22% |

Table 4: Pass@1 and majority@64 performance of different pretraining data mixtures on the AIME 2022-2024 benchmark both before and after doing PPO on GSM8K.

| Pretraining Data Mixture | Pass@1 Base | Pass@1 FT | Maj@64 Base | Maj@64 FT |
|---|---|---|---|---|
| TinyGSM + 4xOMI1 | 0.00% | 0.00% | 0.00% | 0.00% |
| TinyGSM + OMI2 | 2.47% | 6.54% | 6.43% | 13.93% |
| OMI2 + MMQA | 2.89% | 7.93% | 7.40% | 14.36% |
| TinyGSM | 0.00% | 0.21% | 0.21% | 0.75% |
| TinyGSM + OMI1 + OMI2 | 2.47% | 7.18% | 6.54% | 13.50% |

Table 5: Pass@1 and majority@64 performance of different pretraining data mixtures on the AIME 1983-2024 benchmark both before and after doing PPO on GSM8K.

| Pretraining Data Mixture | 1983–2024 Pass@64 Base | 1983–2024 Pass@64 FT |
|---|---|---|
| TinyGSM + 4xOMI1 | 0.00% | 0.00% |
| TinyGSM + OMI2 | 26.37% | 37.41% |
| OMI2 + MMQA | 26.58% | 36.55% |
| TinyGSM | 4.72% | 7.07% |
| TinyGSM + OMI1 + OMI2 | 25.08% | 38.16% |

| Pretraining Data Mixture | 2022–2024 Pass@64 Base | 2022–2024 Pass@64 FT |
|---|---|---|
| TinyGSM + 4xOMI1 | 0.00% | 4.44% |
| TinyGSM + OMI2 | 0.00% | 18.89% |
| OMI2 + MMQA | 8.89% | 21.11% |
| TinyGSM | 1.11% | 8.89% |
| TinyGSM + OMI1 + OMI2 | 11.11% | 15.56% |

Table 6: Pass@64 results on the AIME 1983–2024 and 2022–2024 benchmarks after PPO on GSM8K.

### H.3 Example Generations on MATH and AIME 2022-2024

Below we provide an example problem, an incorrect solution generated by the pretrained 1B model, and a subsequently correct solution generated by the fine-tuned 1B model for both MATH-500 and AIME 2024.

> **Example Problem from MATH-500**
>
> **Problem:** The two-digit number "$B6$," where $B$ is the tens digit, is the square of a positive integer. How many distinct possible values are there for $B$?

> **Example Problem from AIME 2024**
>
> **Problem:** Alice chooses a set $A$ of positive integers. Then Bob lists all finite nonempty sets $B$ of positive integers with the property that the maximum element of $B$ belongs to $A$. Bob's list has 2024 sets. Find the sum of the elements of $A$.

---

### Example Incorrect Generation from MATH-500

Let $B$ be a two-digit number, so $B = 10a + b$, where $a$ is the tens digit and $b$ is the units digit.
Since $B$ is the square of a positive integer, we can write:

$$B = x^2$$

where $x$ is a positive integer.
We can rewrite the equation as:

$$10a + b = x^2$$

Since $a$ and $b$ are digits, we know that $1 \leq a \leq 9$ and $0 \leq b \leq 9$.
We can try different values of $x$ to find the possible values of $B$.
For $x = 1$, we have:
$$10a + b = 1^2 = 1 \Rightarrow b = 1 - 10a$$
$$0 \leq 1 - 10a \leq 9 \Rightarrow 0 \leq 10a \leq 8 \Rightarrow 0 \leq a \leq 0.8$$
This is a contradiction, so $x \neq 1$.
For $x = 2$, we have:
$$10a + b = 2^2 = 4 \Rightarrow b = 4 - 10a$$
$$0 \leq 4 - 10a \leq 9 \Rightarrow 0 \leq 10a \leq 4 \Rightarrow 0 \leq a \leq 0.4$$
This is a contradiction, so $x \neq 2$.
For $x = 3$, we have:
$$10a + b = 3^2 = 9 \Rightarrow b = 9 - 10a$$
$$0 \leq 9 - 10a \leq 9 \Rightarrow 0 \leq 10a \leq 9 \Rightarrow 0 \leq a \leq 0.9$$
This is a contradiction, so $x \neq 3$.
For $x = 4$, we have:
$$10a + b = 4^2 = 16 \Rightarrow b = 16 - 10a$$
$$0 \leq 16 - 10a \leq 9 \Rightarrow 0 \leq 10a \leq 7 \Rightarrow 0 \leq a \leq 0.7$$
This is a contradiction, so $x \neq 4$.
For $x = 5$, we have:
$$10a + b = 5^2 = 25 \Rightarrow b = 25 - 10a$$
$$0 \leq 25 - 10a \leq 9 \Rightarrow 0 \leq 10a \leq 16 \Rightarrow 0 \leq a \leq 1.6$$
This is a contradiction, so $x \neq 5$.
For $x = 6$, we have:
$$10a + b = 6^2 = 36 \Rightarrow b = 36 - 10a$$
$$0 \leq 36 - 10a \leq 9 \Rightarrow 0 \leq 10a \leq 27 \Rightarrow 0 \leq a \leq 2.7$$
This is a contradiction, so $x \neq 6$.
For $x = 7$, we have:
$$10a + b = 7^2 = 49 \Rightarrow b = 49 - 10a$$
$$0 \leq 49 - 10a \leq 9 \Rightarrow 0 \leq 10a \leq 40 \Rightarrow 0 \leq a \leq 4$$
This is a contradiction, so $x \neq 7$.
For $x = 8$, we have:
$$10a + b = 8^2 = 64 \Rightarrow b = 64 - 10a$$
$$0 \leq 64 - 10a \leq 9 \Rightarrow 0 \leq 10a \leq 55 \Rightarrow 0 \leq a \leq 5.5$$
This is a contradiction, so $x \neq 8$.
For $x = 9$, we have:
$$10a + b = 9^2 = 81 \Rightarrow b = 81 - 10a$$
$$0 \leq 81 - 10a \leq 9 \Rightarrow 0 \leq 10a \leq 72 \Rightarrow 0 \leq a \leq 7.2$$
This is a contradiction, so $x \neq 9$.
Therefore, there are no possible values for $B$.
The answer is:

$$\boxed{0}$$

### Example Correct Generation from MATH-500

We are given that the two-digit number $B6$ is the square of a positive integer. Let's denote this integer as $x$. Then we have:

$$B6 = x^2$$

Since $B$ is the tens digit, it can range from 1 to 9.
We can write the equation as:

$$10B + 6 = x^2$$

We need to find the possible values of $B$ that satisfy this equation.
Notice that $x^2$ must be a two-digit number, so $10 \leq x^2 < 100$. This means $4 \leq x < 10$.
We can try different values of $x$ and see if we get a valid $B$:
For $x = 4$, we have $10B + 6 = 16$, so $10B = 10$, which gives $B = 1$.
For $x = 5$, we have $10B + 6 = 25$, so $10B = 19$, which does not give an integer value for $B$.
For $x = 6$, we have $10B + 6 = 36$, so $10B = 30$, which gives $B = 3$.
For $x = 7$, we have $10B + 6 = 49$, so $10B = 43$, which does not give an integer value for $B$.
For $x = 8$, we have $10B + 6 = 64$, so $10B = 58$, which does not give an integer value for $B$.
For $x = 9$, we have $10B + 6 = 81$, so $10B = 75$, which does not give an integer value for $B$.
We found two possible values for $B$: 1 and 3.
The answer is:

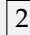

$\boxed{2}$

---

### Example Incorrect Generation from AIME 2024

Let's denote the elements of $A$ as $a_1, a_2, \ldots, a_n$. We want to find the sum of the elements of $A$.

First, we notice that for each $i$ from 1 to $n$, there are at most 2024 distinct sets $B_i$ with the property that the maximum element of $B_i$ belongs to $A_i$.

Since there are 2024 sets in total, we have:

$$\sum_{i=1}^{n} \sum_{b \in B_i} b \leq 2024$$

Now, let's consider the sum of the elements of $A$. We can rewrite it as:

$$\sum_{i=1}^{n} \sum_{b \in B_i} b = \sum_{i=1}^{n} \sum_{b \in A_i} b$$

Since $A_i$ is a subset of $B_i$, we have:

$$\sum_{i=1}^{n} \sum_{b \in A_i} b = \sum_{i=1}^{n} \sum_{b \in A_i} a_i$$

Notice that for each $i$, the sum of the elements of $A_i$ is equal to the sum of the elements of $B_i$ minus the maximum element of $B_i$. Since the maximum element of $B_i$ belongs to $A_i$, we have:

$$\sum_{i=1}^{n} \sum_{b \in A_i} a_i = \sum_{i=1}^{n} \sum_{b \in B_i} a_i - \sum_{b \in A_i} a_i$$

Simplifying, we get:

$$2 \sum_{i=1}^{n} \sum_{b \in A_i} a_i = 2024$$

Since each $a_i$ appears at most 2024 times in the sum, we have:

$$\sum_{i=1}^{n} \sum_{b \in A_i} a_i \leq 2024 \cdot 2 = 4048$$

We also know:

$$\sum_{i=1}^{n} \sum_{b \in B_i} a_i \leq 2024 \cdot \sum_{i=1}^{n} \sum_{b \in A_i} a_i \leq 2024 \cdot 4048$$

Thus:

$$\sum_{i=1}^{n} \sum_{b \in A_i} a_i = 2024 \cdot 2024$$

Then:

$$\sum_{i=1}^{n} a_i = 2024 \cdot n = 2024 \cdot 2024 = \boxed{4096}$$

> **Example Correct Generation from AIME 2024**
>
> Let the elements of $A$ be $a_1, a_2, \ldots, a_n$. A set $B$ is allowed if its maximum element is in $A$.
> Every subset $B$ must be a nonempty set of positive integers such that $\max(B) \in A$.
> Now, fix an element $a \in A$. How many subsets $B$ have maximum $a$? Every such $B$ is a nonempty subset of $\{1, 2, ..., a\}$ with maximum $a$, which means the rest of the elements in $B$ must be from $\{1, 2, ..., a - 1\}$. There are $2^{a-1}$ such subsets (each subset of $\{1, ..., a - 1\}$ plus $a$).
> So the total number of such sets $B$ is:
> $$\sum_{a \in A} 2^{a-1} = 2024$$
>
> Now we want to find a set $A$ of positive integers such that $\sum_{a \in A} 2^{a-1} = 2024$.
> Note that 2024 in binary is:
> $$2024 = 2^{10} - 1 = 1023 \text{ is too small}, \quad 2^{11} = 2048 \text{ is too big}$$
>
> Try breaking 2024 into powers of 2:
> $$2024 = 1024 + 512 + 256 + 128 + 64 + 32 + 8 \Rightarrow A = \{11, 10, 9, 8, 7, 6, 4\}$$
>
> Then the sum of elements of $A$ is:
> $$11 + 10 + 9 + 8 + 7 + 6 + 4 = \boxed{55}$$

## I  RL Fine-tuning on MATH

The results presented in Sections 3 and 4 focus on performing PPO fine-tuning with problems from the GSM8K train split; we also ran PPO with problems from the MATH train split for three of our 1B models pretrained with difference mixtures. Due to computational resources we keep the same hyperparameters as detailed in Appendix C.

We show the change in performance on MATH-500 in Table 7 as well as performance on AIME 1983-2024 and AIME 2022-2024 in Table 9, Table 8, and Table 10. Compared to fine-tuning on GSM8K train questions, we observe less improvements in performance on MATH-500 and similar results when evaluating on AIME, where only pass@64 performance yields significant improvements.

| Pretraining Data Mixture | MATH Pass@1 Base | MATH Pass@1 FT |
|---|---|---|
| TinyGSM + OMI2 | 33.40% | 39.80% |
| OMI2 + MMQA | 34.60% | 42.80% |
| TinyGSM + OMI1 + OMI2 | 33.40% | 39.20% |

| Pretraining Data Mixture | MATH Maj@64 Base | MATH Maj@64 FT |
|---|---|---|
| TinyGSM + OMI2 | 46.20% | 49.20% |
| OMI2 + MMQA | 51.20% | 50.00% |
| TinyGSM + OMI1 + OMI2 | 48.60% | 49.40% |

| Pretraining Data Mixture | MATH Pass@64 Base | MATH Pass@64 FT |
|---|---|---|
| TinyGSM + OMI2 | 80.40% | 83.00% |
| OMI2 + MMQA | 80.60% | 83.80% |
| TinyGSM + OMI1 + OMI2 | 83.40% | 82.40% |

Table 7: Pass@1, majority@64, and pass@64 performance of different pretraining data mixtures on the MATH-500 benchmark both before and after doing PPO on MATH.

| Pretraining Data Mixture | Pass@1 Base | Pass@1 FT | Maj@64 Base | Maj@64 FT |
|---|---|---|---|---|
| TinyGSM + OMI2 | 1.11% | 3.33% | 1.11% | 3.33% |
| OMI2 + MMQA | 0.00% | 1.11% | 0.00% | 2.22% |
| TinyGSM + OMI1 + OMI2 | 0.00% | 2.22% | 1.11% | 3.33% |

Table 8: Pass@1 and majority@64 performance of different pretraining data mixtures on the AIME 2022–2024 benchmark both before and after doing PPO on MATH.

| Pretraining Data Mixture | Pass@1 Base | Pass@1 FT | Maj@64 Base | Maj@64 FT |
|---|---|---|---|---|
| TinyGSM + OMI2 | 2.47% | 6.65% | 6.43% | 11.79% |
| OMI2 + MMQA | 2.89% | 7.72% | 7.40% | 13.40% |
| TinyGSM + OMI1 + OMI2 | 2.47% | 7.82% | 6.54% | 14.36% |

Table 9: Pass@1 and majority@64 performance of different pretraining data mixtures on the AIME 1983–2024 benchmark both before and after doing PPO on MATH.

| Pretraining Data Mixture | 1983–2024 Pass@64 Base | 1983–2024 Pass@64 FT |
|---|---|---|
| TinyGSM + OMI2 | 26.37% | 34.51% |
| OMI2 + MMQA | 26.58% | 34.41% |
| TinyGSM + OMI1 + OMI2 | 25.08% | 35.58% |

| Pretraining Data Mixture | 2022–2024 Pass@64 Base | 2022–2024 Pass@64 FT |
|---|---|---|
| TinyGSM + OMI2 | 10.00% | 18.89% |
| OMI2 + MMQA | 0.00% | 15.56% |
| TinyGSM + OMI1 + OMI2 | 10.00% | 18.89% |

Table 10: Pass@64 results on the AIME 1983–2024 and 2022–2024 benchmarks after PPO on MATH.

