# OpenReview forum: "Echo Chamber: RL Post-training Amplifies Behaviors Learned in Pretraining"
_colmweb.org/COLM/2025/Conference — COLM 2025_

### Official Review · Reviewer_9LBw · 2025-05-04

**Rating:** 7
**Confidence:** 4
**Ethics Flag:** 1

**Summary:**

This paper investigates how RL fine-tuning interacts with the pretraining data of language models in mathematical reasoning tasks. The authors perform a comprehensive, end-to-end analysis using fully open pretraining datasets and various RL algorithms (PPO, GRPO, Expert Iteration) across several model scales. They find that RL post-training amplifies a single distribution from the pretraining mixture, reducing diversity. The favored format is often the most performant one, but it varies with model scale: smaller models favor simpler, code-like formats while larger models shift towards natural language outputs. The paper is well written and easy to follow.

**Questions To Authors:**

- I find the phenomenon demonstrated in this paper, i.e., RL tuning lets the model consistently converge to a dominant output distribution coinciding with a significant increase in performance, is quite similar to those reported in [1] on off-policy DPO. Would the theoretical framework provided in [1] explain the findings in this paper?
- The results show that the converged type or distribution is not directly correlated to the initial ratio of the pretrained data. Would the average initial confidence in different types of responses be a good measurement?
- Some small typos (or inconsistencies), e.g., “150m model” in the caption of Figure 3, should it be “150M”?

[1] Ren, Yi, and Danica J. Sutherland. "Learning dynamics of llm finetuning." ICLR 2025

**Reasons To Accept:**

- Novelty and Relevance: The paper fills a significant gap in understanding how RL interacts with pretraining, especially in controlled settings. This is a good contribution given the opacity in the current state-of-the-art LLMs.
- Clear Empirical Findings: The “echo chamber” effect (convergence to one pretraining format) is clearly and convincingly demonstrated.
- Well-designed experiments: I like the comparison between pass 1 and pass 64 design, which provides us a good overview of how the model’s distribution on these responses looks like (peakier distribution yields higher pass 1 while higher pass 64 means a larger and better support).

**Reasons To Reject:**

- Some of the interesting findings could be further analyzed and explained, e.g., the role of the KL term.
- Only tracking the accuracy and percentage might not explain the phenomenon here. I suggest tracking some confidence-related metrics, e.g., entropy of the distribution, log-probabilities of them, etc., would make the paper stronger.
- There’s no theoretical or mechanistic explanation for why the model collapses to one format. Could this be tied to reward sparsity, easier credit assignment, squeezing effect (mentioned later), or inductive bias?

---

> ### Author Response · Authors · 2025-05-31
>
> We thank the reviewer for their thoughtful review and appreciate their positive comments about the novelty and relevance of our work. We address each of their comments and questions below.
>
> ```Some of the interesting findings could be further analyzed, e.g., the role of the KL term.```
>
> We agree that further investigation of the KL term would be interesting; we have further results in Appendix D (Figure 8 and 9) where we ablated over KL terms and see the same findings in the main paper across different mixtures. For KL = 0, we see slightly more aggressive updates compared to KL = 1e-3.
>
> ```I suggest tracking some confidence-related metrics, e.g., ...```
>
> We’d like to clarify that all of our percentage plots in the paper are averaged across 64 generations (temperature 0.7, top_p 0.95) per question – we will add this in the revision. Thus the empirical frequency of the format occurring is validated across many samples. Following the reviewer’s suggestion, for three of our data mixtures, we plotted the average probability of `def simple_math_problem()` and `Let’s solve this problem using Python code. <llm-code>` occurring after each problem in GSM8K test. These are the first few tokens associated with all TinyGSM and OMI1-style outputs as shown in Appendix B (there isn’t a consistent format for OMI2, so we exclude it). From https://postimg.cc/mhL5j9Yq, we can see that the probabilities track the percentage trends in Figure 2, Figure 4(a) and Figure 4(b) respectively, with more of a gradual increase. We also observe that error bars narrow over training. In general, we observed that the average probabilities of the model’s generations increase throughout training, even after converging in output format—suggesting growing model confidence within the dominant distribution (and is consistent with previous observations about reduced diversity). We agree that this provides a more thorough picture of the empirical phenomenon and are happy to add all of these results to the next revision.
>
> ```There’s no theoretical or mechanistic explanation for why the model collapses to one format...```
>
> We agree with the reviewer’s points that more work is needed towards understanding why this phenomenon emerges, both theoretically and empirically. We have an initial theory for why RL collapses to a single format, and we plan to add a more complete theoretical explanation to the revision. In short, we analyze a setting where the model is trained on a mixture of distributions $D = \sum_i \alpha_i D_i$, where each distribution $D_i$ captures a distinct format. Then, assume that the learned model generates an output distribution that is a mixture distribution of the following form: $f(y|x) = \sum_i \alpha_i f_i(y|x)$, where $f_i(y|x)$ is the distribution of $f(y|x)$ conditioned on generating examples “similarly” to the distribution $D_i$. Then, if we consider only the correct generation from the model $f(y|x)$, we can show that, under some assumptions, this induces a new distribution $D^{new} = \sum_i \frac{\alpha_i Acc(f_i)}{\sum_j \alpha_j Acc(f_j)} D_i$, where $Acc(f_i)$ is the accuracy (on the training distribution) of $f_i$. This essentially means that after one step of a simple RL procedure (similar to REINFORCE), the model will generate a distribution that is skewed towards the format that yields higher accuracy. Applying multiple steps of this procedure will quickly converge to the format with highest accuracy. We will include a more rigorous analysis of this intuitive argument in the revision.
>
> ```Would the theoretical framework provided in [1] explain the findings in this paper?```
>
> Thank you for bringing up this interesting reference! We believe that the phenomenon we are seeing is different from the ‘squeezing’ effect observed in off-policy DPO, which is a result of large negative gradients on the ‘rejected’ response that already has a low probability late in training. For our PPO setting, the per-token KL term in the reward can introduce some negative gradients to incorrect answers at the beginning of training, but due to on-policy sampling, the observed phenomenon becomes more driven by the upweighing of the dominant distribution yielding positive reward, since less performant distributions become highly unlikely to be sampled.
>
> ```The results show that the converged type or distribution is not directly correlated to the initial ratio of the pretrained data...```
>
> In most cases, we found that the best-performing distribution at initialization became the converged distribution; the only instances where this was not the case resulted in performance collapse (Figure 4(b)).
>
> ```Some small typos (or inconsistencies)...```
>
> Thank you for spotting these typos, we will be sure to fix them in the revision!
>
> We thank the reviewer again for their constructive comments to strengthen the results of our work. We welcome further discussion and hope these clarifications help inform the reviewer’s evaluation of our work.

---

> > ### Comment · Reviewer_9LBw · 2025-06-06
> >
> > Thanks for the response, which addresses most of my concerns well. The added experiments and theoretical modeling also look interesting. I am looking forward to seeing the new version of this paper. It also makes sense that the self-amplifying mechanism is different from the "squeezing effect". Maybe the theoretical framework provided in [1] is a possible explanation? In short, during RL sampling, the sampled distribution gradually becomes more and more peaky, and hence some modes disappear, which is equivalent to behavior-amplification to some extent. Anyway, given the improvement claimed in this rebuttal, I am happy to increase my evaluation to 7.
> >
> > [1] Shumailov, Ilia, et al. "AI models collapse when trained on recursively generated data." Nature (2024)

---

### Official Review · Reviewer_va6E · 2025-05-10

**Rating:** 6
**Confidence:** 5
**Ethics Flag:** 1

**Summary:**

The paper presents a systematic end-to-end investigation into how reinforcement learning (RL) fine-tuning shapes the behavior of language models trained for mathematical reasoning. The authors pretrain models from scratch using fully open datasets with varying characteristics and then apply several RL algorithms (PPO, GRPO, Expert Iteration). They find that RL fine-tuning causes models to converge towards a single dominant output distribution, typically aligning with the most performant pretraining dataset format. This effect is shown to be scale-dependent, with smaller models favoring code-like outputs and larger models preferring natural language. The study also demonstrates that fine-tuning on simpler tasks (like GSM8K) can yield improvements on harder ones (like MATH), suggesting generalization of reasoning capabilities.

**Reasons To Accept:**

- Controlled Experimental Design: The authors pretrain models from scratch using open datasets, allowing for full transparency and control over training data. Most of the existing work does not ablate the pre-training data factor.

- Findings on Distributional Collapse: The discovery that RL fine-tuning amplifies a dominant pretraining format (i.e., "echo chamber" effect) provides a novel perspective on how RL interacts with model representations, offering valuable implications for interpretability and diversity.

- Scale-Specific Analysis: By comparing 150M and 1B models, the paper reveals scale-dependent biases in format preference and performance, enriching our understanding of model capacity and generalization.

**Reasons To Reject:**

- Format vs. Semantics: One of the core findings is that RL fine-tuning pushes the model toward a dominant output format, but it’s unclear whether this reflects true improvements in reasoning or just better adherence to a surface-level template. Are these models actually reasoning better, or just getting better at mimicking the structure of their most familiar training examples?

- Narrow Dataset Scope: The authors do well to control the pretraining pipeline with open datasets, but the scope is quite limited: centered on mathematical instruction. This is understandable given the focus of the study, but it raises questions about generality. Would similar dynamics emerge in other reasoning and non-reasoning tasks? Could this kind of format collapse still occur if the model were exposed to less structured or more heterogeneous instruction types?

- Heavy Reliance on PPO: PPO is the main algorithm used, and while GRPO and Expert Iteration are mentioned, they’re treated more as side notes. It would be good to understand how much of the observed behavior is an artifact of PPO itself. For instance, would methods with less aggressive policy updates show more format diversity? Or could different RL objectives (like reward shaping with process-level feedback) lead to less homogenization?

---

> ### Author Response · Authors · 2025-05-31
>
> We thank the reviewer for their insightful comments and for taking the time to review our work. We address their points one by one below.
>
> ```Format vs. Semantics: One of the core findings is that RL fine-tuning pushes the model toward a dominant output format, but it’s unclear whether this reflects true improvements in reasoning or just better adherence to a surface-level template. Are these models actually reasoning better, or just getting better at mimicking the structure of their most familiar training examples?```
>
> We thank the review for this point. Importantly, we believe the two aspects—format mimicry and genuine reasoning improvement—are not mutually exclusive. We believe that these models are indeed improving in their reasoning capabilities and we have provided evidence in our work. Concretely, in section 3.3 we show that even when the data mixture contains a single distribution repeated in various proportions (in this case TinyGSM repeated 1x, 2x, 4x and 8x), we get consistent improvement during training (Figure 5), thus showing that when the models are provided with a single output format, they are still able to improve and better adhere to the format. Moreover, in Section 4, we show that we are able to achieve some performance transfer from GSM8K to MATH500 and AIME. The models are able to fix mistakes they were doing before RL, and we categorize these mistakes using a GPT judge in Appendix G, Figures 21 and 22.
>
>
> ```Narrow Dataset Scope: The authors do well to control the pretraining pipeline with open datasets, but the scope is quite limited: centered on mathematical instruction. This is understandable given the focus of the study, but it raises questions about generality. Would similar dynamics emerge in other reasoning and non-reasoning tasks? Could this kind of format collapse still occur if the model were exposed to less structured or more heterogeneous instruction types?```
>
> We agree with the reviewer’s point that more work in this direction is necessary. An immediate direction that could be tested is in coding tasks (for reasoning), for example by seeing if a certain language with less/more semantic complexity is preferred (for example C vs Python). For non-reasoning, one challenge would be finding a way to tag in a similar fashion preference datasets.
> We speculate that a similar dynamic would emerge given the nature of RL optimization: maximizing the expected reward. Thus, if on average a certain distribution gives the bigger expected reward, this distribution will be amplified by the algorithm.
>
> ```Heavy Reliance on PPO: PPO is the main algorithm used, and while GRPO and Expert Iteration are mentioned, they’re treated more as side notes. It would be good to understand how much of the observed behavior is an artifact of PPO itself. For instance, would methods with less aggressive policy updates show more format diversity? Or could different RL objectives (like reward shaping with process-level feedback) lead to less homogenization?```
>
> Algorithm choice was indeed a key factor we aimed to investigate; however, due to space constraints, we report these comparisons in Appendix F. We do observe that less aggressive policy updates maintain a more heterogeneous mixture. We ablate over the KL (between the current policy and initialization, that is typically added as a regularizer in LLM post-training) and find that higher values of KL maintain a more diverse mixture (Appendix D, Figure 8). With regards to GRPO, EI and other policy optimization algorithms, in our experiments (Appendix F.1 for GRPO, and F.2 for EI), we notice a similar upweighing of a specific distribution behaviour as observed in PPO - however due to computational constraints, we haven’t swept the hyperparameters for these methods as much as PPO, leaving this to future work. Similarly, while at the moment we cannot speculate whether reward shaping or PRMs would give rise to different dynamics, we believe that this is an important avenue for future work.
>
> We hope our responses help clarify the points raised and would be glad to continue the discussion. We appreciate the reviewer’s engagement and hope these clarifications provide useful context for their assessment.

---

> > ### Comment · Reviewer_va6E · 2025-06-01
> >
> > The reviewer would like to thank and acknowledge that they read the author's rebuttal, and would like to lean towards the acceptance of the paper.

---

### Official Review · Reviewer_cgzq · 2025-05-10

**Rating:** 8
**Confidence:** 3
**Ethics Flag:** 1

**Summary:**

The paper seeks to explore the mechanism(s) underlying the ability of reinforcement learning fine-tuning to improve performance on mathematical reasoning and coding tasks.  The authors pre-train language models of different scale from _scratch_ on curated mixtures of open datasets.  These datasets have distinct format that function as _fingerprints_ which allow the authors to track the output distributions of the language model before and after reinforcement learning fine-tuning.  They find that language models fine-tuned for mathematical reasoning and coding tasks often converge to the distribution of a single component of the training corpus, and that this specific distribution influenced by size.

**Questions To Authors:**

None

**Reasons To Accept:**

*	The paper systematically establishes a link between pre-training data and a model’s output distribution after RL fine-tuning across three RL algorithms.
*	The experimental setup concise and crystal-clear, with sound rationale for the selected architecture, datasets and reinforcement learning algorithms.
*	Pre-training from _scratch_ on datasets with _fingerprints_ is insightful as it provides a mechanism to track otherwise intractable output distributions.
*	Results obtained from experimenting with the KL divergence coefficient are interesting, and illustrate the coefficient is indeed effective at guiding the output distribution.
*	The observation that pre-training on a single dataset (line 195 -197) is potentially impactful.
*	The finding that smaller models seem to prefer code-style pre-training, while a larger model’s output distribution converges to a natural language distribution is a strong foundation for future research.
*	Results are thoroughly ablated with respect to dataset mixtures and reinforcement earning algorithms.
*	The paradigm of moving towards pre-training from _scratch_ instead of relying on existing pre-trained models is excellent.

**Reasons To Reject:**

*	While the empirical findings are highly valuable, more insight on the underlying _mechanism_ might be helpful.
*	Exploring other families of tasks or model architectures could make the findings more durable.
*	(minor) Presenting results in tabular format would help with comparisons among settings.
*	(minor) Providing examples of the _distinct formats_ which facilitate distribution tracking would make this important consideration less abstract.

---

> ### Author Response · Authors · 2025-05-31
>
> We sincerely thank the reviewer for their very positive comments in appreciation of our work in their "Reasons to Accept" section of the review. We address each of the the points raised by the reviewer below.
>
> ```While the empirical findings are highly valuable, more insight on the underlying mechanism might be helpful.```
>
> We agree with the reviewer’s points that more work is needed towards understanding why this phenomenon emerges, both theoretically and empirically. While we do not have a definitive idea as to the actual mechanism underlying our findings, we speculate that upweighing the “best distribution” from the data mixture is fundamentally how RL operates. Concretely, the main goal of RL is to maximize the expected reward, and since on average generating samples from a specific distribution does this more often than from a worse one one, the first one gets upweighted through optimization.
>
> ```Exploring other families of tasks or model architectures could make the findings more durable.```
>
> In our current setup, we pretrain models based on OLMo (a standard autoregressive language model architecture) from scratch on data mixtures that we generate. However we do agree with the reviewer that extending our experimental setup to other tasks in the verifiable domain setting (such as code), or non-verifiable (such as human preference data), as well as other algorithms like DPO would be an interesting future direction.
>
> ```Presenting results in tabular format would help with comparisons among settings.```
>
> We thank the reviewer for this comment and we will include our main results in a tabular setting in a future version.
>
> ``` Providing examples of the distinct formats which facilitate distribution tracking would make this important consideration less abstract.```
>
> We would like to point the reviewer towards Appendix B (specifically B.1, B.2, B.3) where we provide example samples from each of the datasets that we use in the training mixtures i.e. TinyGSM, OpenMathInstruct1 and OpenMathInstruct2. More specifically, for TinyGSM we can track the tag `def simple_math_problem`, for OMI1 we can track the `<llm-code>` tag, and for OMI2 we can check if neither of the previous tags appear since it only contains natural language).
>
> We are grateful for the reviewer’s encouraging evaluation and hope our clarifications address the remaining concerns. We would be glad to elaborate further during the follow-up discussion period if needed.

---

> > ### Comment · Reviewer_cgzq · 2025-06-08
> >
> > After reading the other reviews and author responses, I reaffirm my rating.  I believe that this paper provides strong inspiration for future work.

---

### Decision · Program_Chairs · 2025-07-08

**Decision:**

Accept

**Comment:**

This paper proposes a controlled study where they analyze the interaction between the pretraining mixture and RL finetuning finding that different mixtures lead to very different output distributions after RL finetuning. This is an interesting finding that all reviewers appreciated. Reviewers were impressed by the effort and clean experimental methodology and findings.

As a sidenote, this finding is perhaps not surprising given that the solution to the problem of maximizing E(R) - KL(\pi || \pi_ref) is the original distribution tilted by an exponent of the reward  (\pi^*(y | x) \propto \pi_ref(y | x) exp(R(x,y / \beta).